# Induction of Angiogenesis by Genetically Modified Human Umbilical Cord Blood Mononuclear Cells

**DOI:** 10.3390/ijms24054396

**Published:** 2023-02-23

**Authors:** Dilara Z. Gatina, Ilnaz M. Gazizov, Margarita N. Zhuravleva, Svetlana S. Arkhipova, Maria A. Golubenko, Marina O. Gomzikova, Ekaterina E. Garanina, Rustem R. Islamov, Albert A. Rizvanov, Ilnur I. Salafutdinov

**Affiliations:** 1Institute of Fundamental Medicine and Biology, Kazan (Volga Region) Federal University, 420008 Kazan, Russia; 2Department of Medical Biology and Genetics, Kazan State Medical University, 420012 Kazan, Russia

**Keywords:** human umbilical cord blood mononuclear cells, angiogenesis, recombinant adenoviruses, gene modification, transgene expression, VEGF, FGF2, SDF1α, NUDE mice, Matrigel plugs, cytokine profile

## Abstract

Stimulating the process of angiogenesis in treating ischemia-related diseases is an urgent task for modern medicine, which can be achieved through the use of different cell types. Umbilical cord blood (UCB) continues to be one of the attractive cell sources for transplantation. The goal of this study was to investigate the role and therapeutic potential of gene-engineered umbilical cord blood mononuclear cells (UCB-MC) as a forward-looking strategy for the activation of angiogenesis. Adenovirus constructs Ad-VEGF, Ad-FGF2, Ad-SDF1α, and Ad-EGFP were synthesized and used for cell modification. UCB-MCs were isolated from UCB and transduced with adenoviral vectors. As part of our in vitro experiments, we evaluated the efficiency of transfection, the expression of recombinant genes, and the secretome profile. Later, we applied an in vivo Matrigel plug assay to assess engineered UCB-MC’s angiogenic potential. We conclude that hUCB-MCs can be efficiently modified simultaneously with several adenoviral vectors. Modified UCB-MCs overexpress recombinant genes and proteins. Genetic modification of cells with recombinant adenoviruses does not affect the profile of secreted pro- and anti-inflammatory cytokines, chemokines, and growth factors, except for an increase in the synthesis of recombinant proteins. hUCB-MCs genetically modified with therapeutic genes induced the formation of new vessels. An increase in the expression of endothelial cells marker (CD31) was revealed, which correlated with the data of visual examination and histological analysis. The present study demonstrates that gene-engineered UCB-MC can be used to stimulate angiogenesis and possibly treat cardiovascular disease and diabetic cardiomyopathy.

## 1. Introduction

Angiogenesis is the growth of new blood vessels from pre-existing vessels, an essential process for development, wound healing, and the restoration of blood flow and oxygen supply to tissues after injury. One of the main tasks of modern medicine is the stimulation of the processes of angiogenesis in the treatment of vascular diseases. To date, many approaches have been proposed for the induction of therapeutic angiogenesis. Among the proposed methods are surgical methods [1], the use of inducer proteins [2], recombinant DNA molecules [3], inducer genes [4], and the use of various cell types [5,6], including ex vivo genetically modified cells [7]. In this aspect, human umbilical cord blood mononuclear cells (UCB-MC) seem to be a promising “tool” for stimulating angiogenesis through the delivery of genetic engineering systems, expression of recombinant proteins, and possibly direct participation in new vessel formation. The choice of UCB-MC in studies for cell and gene-cell therapy looks promising because of some advantages of this cellular source. Umbilical cord blood contains many stem/progenitor cells and can be obtained easily [8]. The mononuclear fraction of UCB contains populations of different immature cells capable of differentiating into many cell types [9]. Cell populations that have been discovered in UCB are hematopoietic stem cells (HSCs), endothelial progenitor cells, mesenchymal stem cells (MSCs), unrestricted somatic stem cells (USSCs), and side population cells [10,11,12,13]. As cellular material for transplantation or carriers for genetic constructs, UCB-MCs have low immunogenicity because they do not express all the antigens on the cell membrane. This feature enhances the ability to cross donor-recipient HLA disparities. It allows for the usage of UCB-MC for transplantation in non-fully compatible HLA recipients with a much lower incidence of grade II-IV acute GVHD (graft versus host disease) cases after transplantation [14,15,16,17]. Furthermore, UCB-MCs can prevent the oncological transformation of recipient cells after transplantation [15]. Another appealing reason for using UCB cells for cell therapy is their ability to produce various biologically active molecules, such as proteins with antioxidant properties, angiogenic, neurotrophic, and growth factors [18,19,20,21,22], which make them suitable for effective stimulation of regenerative processes in non-compatible recipients for a short time before the immune system eliminates them. Overall, UCB cell transplantation can replace dead cells, prevent further death of surviving cells, and stimulate regeneration by secreting biologically active molecules. A genetic modification of UCB cells can enhance their ability to regenerate tissue [23,24]. This approach unites the advantages of cell- and gene therapy. Genetically modified UCB cells can provide targeted delivery of therapeutic genes and expression of recombinant molecules at the regeneration site. For example, our previous studies showed the positive effect of genetically modified umbilical cord blood mononuclear cells (UCB-MC) simultaneously produces three recombinant molecules (vascular endothelial growth factor (VEGF), glial cell-derived neurotrophic factor (GDNF), and neural cell adhesion molecule (NCAM) in animal models of amyotrophic lateral sclerosis [25], spinal cord injury [26] and stroke [7,27]. Many state-of-the-art methods and models for studying angiogenesis have been proposed, which are well analyzed in the review articles [28,29,30]. Among various models, the in vivo angiogenesis plug assay, which uses basement membrane extracts (BME) or Matrigel, is widely used for evaluating pro- or anti-angiogenic factors during in vivo angiogenesis. This assay is reliable, easy to perform without special equipment, reproducible, quantitative, and quick [31,32,33]. Matrigel predominantly contains laminin III, collagen IV, heparan sulfate proteoglycans, and various growth factors. The assay is performed by injecting the liquid Matrigel into the subcutaneous space of an animal at 4 °C, which solidifies to form a plug at body temperature. Over time, blood vessels sprout into the plug. The number of plug sites per animal can be several, allowing multiple test compounds or concentrations to be tested. Thus, drug screening can also be evaluated for effects on the activity of angiogenic or anti-angiogenic factors [34,35,36,37]. The drug can be placed in the plug with the test factor by mixing with the Matrigel matrix or given to the host animal. Cells or exosomes can also be examined when mixed into the gel to produce angiogenic factors. Furthermore, the assay is highly versatile. For example, the role of certain genes can be evaluated using genetically modified mice (overexpressing or ablating a protein gene) or animal models of diseases. This report aimed to study the effect of genetically modified umbilical cord blood mononuclear cells overexpressing recombinant proangiogenic proteins VEGF165, FGF2, and SDF1α on the induction of angiogenesis. Furthermore, we assessed the influence of all three factors on the tone of the secretory profile of modified UCB-MCs and tubule formation in the in vivo Matrigel plug assay. The present study shows that when combined with UCB, the three factors can enhance angiogenesis and be useful for developing new therapies.

## 2. Results

### 2.1. Characterization of Isolated Human UCB-MCs

Isolated cells demonstrated high viability (>97%) and included CD45+ lymphocytes (58.9%). CD45+CD3+ lymphocytes constituted 59.2%, while CD14+ macrophages constituted 7.3%. This ratio of the central populations of blood cells (lymphocytes, T-lymphocytes, and monocytes) is believed to be typical for human UCB-MCs. We also examined the percentage of CD34+ blood cells among isolated UCB-MCs. According to the obtained data, CD34+ cells constituted 0.4% of CD45+ cells. In addition, 91.8% of CD45+CD34+ cells expressed CD38. Furthermore, 90% of the CD45+CD34+ cells had the phenotype CD90+. The flow cytometry results are shown in Figure 1.

Immunophenotyping of a pool of CD34-positive cells showed that genetic modification and expression of recombinant factors by cells did not affect the viability and endothelial cell markers (Figure 2).

### 2.2. Transduction of UC-MCs with Recombinant Adenoviruses Increased Transgene Expression

It has been demonstrated that genetic modification of the UCB-MCs with recombinant adenoviruses (Ad-VEGF, Ad-FGF2, Ad-SDF1α, or Ad-EGFP) did not affect cell viability. Moreover, it has been shown that UCB-MCs transduced with Ad-EGFP exhibited green fluorescence, confirming the efficiency of transduction (Figure 3A). Furthermore, EGFP expression was sustained for 30 days after a genetic modification of UCB-MCs. According to the flow cytometry results, EGFP+ cells constituted 28 ± 2.7% (Figure 3B). Analysis of the mRNA expression of VEGF165, FGF2, and SDF1α in genetically modified human UCB-MCs was carried out using qPCR. It has been established that genetic modification of hUCB-MCs with Ad-VEGF165 results in augmented VEGF expression (190.6 ± 8.9 fold). Simultaneous transduction with Ad-VEGF165, Ad-FGF2, and Ad-SDF1α resulted in the upregulation of VEGF, FGF2, and SDF1α expression (198.6 ± 0.45; 204.2 ± 0.36 and 140.9 ± 0.32 fold respectively) compared to non-transfected cells, and cells modified with Ad-EGFP (Figure 3C). The obtained results are evident for efficient modification of hUCB-MCs with developed genetic constructs which provide a synthesis of target genes in vitro.

### 2.3. Genetically Modified hUCB-MCs Produce a Broad Range of Cytokines, Chemokines, and Growth Factors

A complete analysis of all cytokines and chemokines measured in the Luminex assays demonstrated that gene modification and gene expression did not change levels of multiple anti and proinflammatory cytokines as well as chemokines. The results obtained from the eight donors in comparison to the untreated control are shown in Table 1 (Appendix A). We have not observed any statistically significant differences in cytokine and chemokine secretion between the groups of non-transfected cells and genetically modified ones except for upregulated levels of recombinant proteins in corresponding groups. Multiplex analysis revealed statistically significant (*p* < 0.05) upregulation of VEGF secretion (1087.12 ± 169.11 pg/mL) in UCB-MCs modified with Ad-VEGF compared to the UCB-MCs treated with Ad-EGFP (52.31 ± 10.36 pg/mL) and non-treated cells (51.75 ± 8.65 pg/mL). Simultaneous transduction with Ad-VEGF, Ad-FGF2, and Ad-SDF1 has resulted in the increased production of VEGF (701.94 ± 96.99 pg/mL), FGF2 (576.27 ± 57.83 pg/mL), and SDF1α (622.39 ± 113.07 pg/mL) (Figure 3D). Obtained results correlate with the data presented above of RT-qPCR and confirm the capacity of recombinant adenoviruses for infection of target cells.

It is also worth emphasizing that the UCB-MC-VEGF-FGF2-SDF1 and UCB-MC-VEGF did not differ from UCB-MC-EGFP and UCB-MC-NTC in vitro studies. What can be seen from the data of morphological, and phenotypic studies are the profiles of secreted factors. Therefore, UCB-MC-EGFP is the ideal control in our study in vivo.

### 2.4. Transplantation of Genetically Modified Cells Promotes Angiogenesis In Vivo

Matrigel mixtures were implanted into the subcutaneous space of the dorsal region in mice after seven days post-transplantation when implanted Matrigel samples containing genetically modified UCB-MCs were extracted from Balb/c nude mice. Embedded fragments represented discs with d = 10 mm and 2 mm height. The color of the implants correlated with vascularization density. The color of the implants varied from milky-white (Matrigel without cells and Matrigel with UCB-MC + Ad-EGFP) to red-brown (Matrigel with UCB-MC + Ad-VEGF165 + Ad-FGF2 + Ad-SDF1α) which is due to the vascular formation and presence of blood cells, particularly, erythrocytes (Figure 4A). Gross histological hematoxylin and eosin (H&E) staining of extracted plugs showed the absence of inflammatory sites. The skin and subcutaneous tissue in the area of implantation were not visually changed (Figure 4B). We have established that in isolated subcutaneous implants containing hUCB-MC, human-transduced Ad-VEGF165, or a combination of Ad-VEGF165, Ad-SDF1α, and Ad-FGF2, the hemoglobin concentration was significantly higher in comparison with Matrigel fragments without cell administration and implants with UCB-MCs transduced Ad-EGFP. Moreover, the significantly higher concentration of hemoglobin was determined in the samples containing UCB-MCs transduced with Ad-VEGF165, Ad-SDF1α, and Ad-FGF2 compared to the group with UCB-MCs transduced with single Ad-VEGF165 (Figure 4C). Moreover, we observed a two-fold increase of mCD31 mRNA expression in plugs containing hUCB-MC transduced Ad-VEGF165 or a combination of Ad-VEGF165, Ad-SDF1α, and Ad-FGF2 compared to controls. Moreover, we did not discover the difference between Ad-VEGF165 and the group with a mixture of Ad-VEGF165, Ad-SDF1α, and Ad-FGF2.

Analysis of the mRNA expression of VEGF165, FGF2, and SDF1α in genetically modified UCB-MCs in Matrigel implants was evaluated by RT-qPCR. Notably, obtained results confirmed the expression of target genes in genetically modified UCB-MCs implanted in Matrigel even at one-week post-transplantation. We discovered that the Matrigel complexes containing UCB-MC Ad-VEGF gave rise to more abundant VEGF mRNA than UCB-MC Ad-EGFP and PBS (Matrigel samples without UCB-MCs). Likewise, UCB-MCs contemporaneously transduced with Ad-VEGF, Ad-FGF2, and Ad-SDF1α exhibited upregulated levels of mRNA expression of VEGF, FGF2, and SDF1α. (Figure 5A). During histological analysis of implants, it has been shown that control—PBS (Matrigel samples without UCB-MCs) contained small amounts of migrated fibroblast-like cells. Visually, the implants were surrounded by a thin connective tissue capsule, which contained rare capillaries in a density of 1.5 ± 0.5 units/mm^2^. In samples with implanted UCB-MCs transduced with a cocktail of adenoviruses (Ad-VEGF165, Ad-FGF2, and AdSDF1α), Matrigel mass contained a residual amount of VEGF+ cells. These vessels localized close to the capsule and migrated fibroblasts, some of which were positive for SDF1α and FGF2. In Matrigel samples with implanted UCB-MCs genetically modified with Ad-EGFP, we found single and small rounded clusters of EGFP-positive cells and rare migrated fibroblast-like cells. The implants were surrounded by a thin connective tissue capsule, from which strands of connective tissue grew into its depth with capillaries found in a density of 7.5 ± 3 units/mm^2^. Vessels were located close to the capsule. Fibroblasts that migrated into Matrigel expressed SDF1α and FGF2. Expression of VEGF, FGF2, and SDF1α in the implanted UCB-MCs were not confirmed. In the group with UCB-MCs modified with Ad-VEGF165, implant samples presented Matrigel mass with single small, rounded clusters of VEGF-positive UCB-MCs cells and rare migrated fibroblast-like cells. The implants were surrounded by a thin connective tissue capsule, from which strands of connective tissue grew more profound into the central regions of the implant with capillaries‘ density of 16 ± 5 units/mm^2^. In the group of UCB-MCs simultaneously transduced with a combination of Ad-VEGF165, Ad-SDF1α and Ad-FGF2, implant samples were represented by the mass of Matrigel with single and small rounded clusters of UCB-MCs, as well as rare migrated fibroblast-like cells. The implant was surrounded by a thin connective tissue capsule, from which the connective tissue and vessels of various calibers grew to the center of the implant with a capillary density of 23 ± 5 units/mm^2^. Implanted UCB-MCs expressed VEGF, FGF2, and SDF1α (Figure 5B).

## 3. Discussion

Adenoviruses mediate gene transfer into dividing and quiescent cells and can be produced with a significant titer. The high immunogenicity of adenoviruses as vehicles for the delivery of therapeutic genes represents one of the main disadvantages resulting in the activation of the immune response in immune-competent organisms and the absence of expression of the target therapeutic genes [38]. However, this negative effect is eliminated when using an ex vivo gene therapy approach. Moreover, adenoviral systems promote transient transgene expression due to their non-integration into the host cell genome [39]. However, this negative point might become beneficial for gene therapy based on growth factors: induction of angiogenesis does not require the prolonged expression of therapeutic proteins but, more importantly, their synergistic effect [40]. The absence of integration of adenoviruses eliminates the risk of insertional mutagenesis, which is a typical problem when using retroviral vectors [39]. Adenoviral vectors demonstrate comparatively low efficiency of genetic modification of hematopoietic cells, which might be increased with a higher concentration of virus [41] or its specific treatment, resulting in augmented tropism [42]. In the present study, we chose the adenovirus delivery vectors containing VEGF, FGF2, and SDF1α to investigate the angiogenic effect of UCB-MC in vitro and the Matrigel plug assay in Nude mice. In our investigation, cellular carriers expressed phenotype typical for UCB-MCs, and about 30% of the cells were efficiently transduced with an MOI of 10. The transduction efficiency correlated with previous results and other research groups’ data [42,43]. After in vitro transduction, the UCB-MCs expressed the recombinant mRNA of proangiogenic factors in the cytoplasm and secreted those factors into the culture medium, which in our study we confirmed by RT-qPCR and immunological studies. The obtained data correlates with our previously published results [7].

Various approaches were proposed for stimulating therapeutic angiogenesis based on the delivery methods of genetically engineered systems expressing a broad range of proangiogenic factors. The therapeutic efficacy of proangiogenic factors has been proven in numerous experiments on animal models [44] and in several clinical studies [45,46]. The key inducers of angiogenesis, VEGF, FGF2, EGF, SDF1α, and PDGF-BB, are most often used as genetic components [2]. In particular, VEGF is perhaps the most characterized and frequently used mitogen in creating gene therapy systems and in the induction of therapeutic angiogenesis. VEGF is a crucial participant in forming new blood vessels and can induce the growth of pre-existing ones [47]. However, Zentilin et al. reported that overexpression of VEGF induced leaky neovessels that missed connecting correctly with existing vessels [48]. The FGF family includes vertebrates’ most versatile growth factors that play critical roles in many biological processes, including angiogenesis [49]. FGF, similar to VEGF, is a pleiotropic molecule capable of acting on various cell types, including endodermal, mesenchymal, and neuroectodermal origin cells. It has been shown that FGF2 induces the expression of VEGF and several other factors by endothelial cells through autocrine and paracrine mechanisms [50,51]. SDF1α is a constitutively expressed and inducible chemokine, associated with various physiological and pathological processes, including embryonic development, homeostasis maintenance, and angiogenesis activation [52]. There is evidence that the administration of SDF1α increases blood flow and perfusion via the recruitment of endothelial progenitor cells (EPCs). SDF1α binds exclusively to CXCR4 and has CXCR4 as its only receptor [53]. Compared with the effects of other angiogenic growth factors, SDF1α has unique properties. The generation of hyperpermeable vessels, a significant characteristic of VEGF-stimulated angiogenesis, may not be observed after injection of SDF1α contributes to the stabilization of neovessel formation by recruiting CXCR4 + PDGFR+ cKit+ smooth muscle progenitor cells during recovery from vascular injury [54]. Extensive evidence suggests that SDF1α up-regulates VEGF synthesis in several cell types, whereas VEGF and basic FGF induce SDF1α and its receptor CXCR4 in endothelial cells [55]. However, it should also be noted that in a wide range of studies using various models, the mutual synergistic role of VEGF, FGF2, SDF1α, and countless other factors responsible for the formation of vessels has been shown [56,57,58,59]. It is generally known that an optional cellular source for allogenic transplantation should meet the following criteria: it must be less immunogenic and contain a sufficient amount of immature cells capable of differentiation in various directions; it should have a prolonged period of storage and potency for expansion. Most gene-cell-mediated therapy protocols intend genetic modification of target cells with different vectors, providing stable expression of target proteins. Human UCB-MCs might be easily isolated and characterized; these cells exhibit low immunogenicity and are composed of unique populations of progenitor cells capable of differentiation into endothelial, muscular, and neural cells, etc. Mononuclear cells from umbilical cord blood are a well-characterized group of cells that are extensively used in pre-clinical and clinical trials of therapy for various human diseases and the induction of therapeutic angiogenesis as well [60]. However, relatively small amounts of UCB-MCs for achieving sufficient therapeutic effect remain the main limitation for its extensive introduction in the clinic [61]. To increase its biological activity, it was proposed to mix different cell pools with further genetic modification [62]. Contemporary cell-mediated approaches to gene therapy suggest UCB-MC as a cell carrier for the delivery of various therapeutic genes. This concept assumes either the differentiation of transplanted cells into different cell types or the realization of therapeutic effects due to the secretion of a broad range of bioactive molecules [63]. Furthermore, our previous study has demonstrated that UCB-MCs are capable of transferring therapeutic genes and promoting evident therapeutic effects using different models, such as ALS [64], SCI [25,26], and stroke [27]. Similar results were obtained in investigations dedicated to therapies for hematologic and non-hematologic disorders [65,66,67,68,69]. At the same time, there is no current data about the influence of the simultaneous transduction of several recombinant adenoviruses on the secretome profile and angiogenic properties of modified hUCB-MCs. A sustained balance of proangiogenic factors and their synergetic effect is essential for functional vascular formation. In the present study, we developed the UCB-MC application to simultaneously deliver many genes (VEGF, FGF2, and SDF1α) to stimulate angiogenesis. Our previous studies also showed the approach of preventive gene therapy with many genes to positively affect stroke. Adenoviral vectors carrying genes encoding vascular VEGF, glial cell-derived neurotrophic factor (GDNF), and NCAM or gene-engineered umbilical cord blood mononuclear cells (UCB-MC) overexpressing recombinant proteins were intrathecally injected before distal occlusion of the middle cerebral artery in the rat. Morphometric and immunofluorescence analysis revealed a reduction in infarction volume and a lower number of apoptotic cells. It decreased the expression of Hsp70 in the peri-infarct region in gene-treated animals [7].

The heterogeneous cell population from the mononuclear fractions UCB-MCs secretes different anti-inflammatory, pro-inflammatory cytokines, chemokines end grow factors [70]. Previously, it was shown that the duration of cultivation, cultivation medium, and the additives used in the culture are the main factors influencing the production of cytokines by UCB-MCs. Our study describes the profile of cytokines and chemokines released by UCB-MC following their in vitro gene modification by adenoviruses. Five groups of secreted factors were investigated: pro-inflammatory cytokines (IL-6, IL-1β, and TNF), an anti-inflammatory cytokine (IL-4 and IL-10), TH1-type cytokines (IL-12 and IFN-γ), chemokines (IL-8, MIP-1α, MIP-1β, and MCP-1) and growth factors (VEGF, FGF2, and SDF1α). Interestingly, the range of cytokine, chemokine, and growth factor concentrations detected in the supernatants of UBC-MC varied between donors, indicating major individual heterogeneity, comparable with previously published data [71].

The highest secretion level by modified and unmodified cells was shown for IL-8 and MCP-1. These factors are known to be produced more intensively than any other chemokines in the human body and are seen as the first line of defense in inflammatory responses [72]. In addition, the cells also secreted high concentrations of GROα, IL-6, MIF, MIP-1α, MIP-1β, and SCGF-β. Unfortunately, adenoviruses are potent activators of the innate and adaptive immune systems. The administration of high doses of Ad-based vectors to animals or patients, primarily through the intravascular pathway, leads to severe immunopathology manifested by cytokine storm syndrome, disseminated intravascular coagulation, thrombocytopenia, and hepatotoxicity, which can lead to morbidity and also death [69]. Research by Teigler et al. on peripheral blood mononuclear cells (PBMCs) showed that their stimulation with the Ad vector increases the secretion of IFN-γ, INF2α, IL-15, G-CSF, MIG, and IP-10. Supporting this perspective, it is worth emphasizing that the study’s authors used 10^3^ vp/cell [73]. Previous studies have shown that treatment of myeloid dendritic cells and plasmacytoid dendritic cells with Ad5 does not lead to an increase in IFN production by them, even at the highest exposed rAd (100 vp/cell) [43]. Our previous examination has shown that genetic modification UCB-MC and expression of transgenes VEGF or EGFP did not influence the global transcriptome landscape [74]. In this study, we demonstrate that a gene-cell system with simultaneous delivery of genes based on UCB-MC can generate the expression of several transgenes both in vitro and in vivo. Furthermore, the UCB-MC-VEGF165 and UCB-MC-VEGF-FGF2-SDF1α Matrigel plugs in mice were filled with red blood cells and showed vessel-like structure formation. We did not find significant differences between the UCB-MC-VEGF and UCB-MC-VEGF-FGF2-SDF1α groups in the present study. Although in line with the RT-qPCR data and immunology tests, levels of expression of VEGF, SDF1α, and FGF varied. Perhaps this is because we used a small amount of cellular material and a short exposure period to Matrigel fragments.

Furthermore, the UCB-MC-VEGF165 and UCB-MC-VEGF-FGF2-SDF1α Matrigel plugs in mice were filled with red blood cells and showed vessel-like structure formation. We did not find significant differences between the UCB-MC-VEGF165 and UCB-MC-VEGF-FGF2-SDF1α groups in the present study. Although in line with the RT-qPCR data and immunology tests, levels of expression of VEGF, SDF1α, and FGF varied. Perhaps this is because we used a small amount of cellular material and a short exposure period to Matrigel fragments.

## 4. Materials and Methods

### 4.1. Obtaining Recombinant Adenovirus Ad-SDF1α

The creation of expression constructs based on adenovirus was carried out by using molecular cloning methods of Gateway-cloning technology (Invitrogen), as described previously [75]. Briefly, subcloning of SDF1α from the plasmid vector pBud-VEGF-SDF1α into the intermediate vector pDONR221 was performed [76].

### 4.2. The Production of Recombinant Adenoviruses

The HEK293A cells were infected with a coarse viral runoff to prepare the necessary amounts of Ad-VEGF, Ad-FGF2, Ad-SDF1α, and Ad-EGFP adenoviruses. To purify viral particles from cell debris, supernatants were filtered through 0.22 µm filters and centrifuged in a gradient of cesium chloride. Virus dialysis was performed using a membrane with a pore throughput of 3.5 kDa in two stages. After purification and concentration, the resulting recombinant adenoviruses were titrated by optical density, as well as by plaque formation. The titer of the recombinant adenoviruses we obtained was from 1 to 3.8 × 10^9^ PFU/mL. The viral titer values were guided by the genetic modification of human UCB-MC.

### 4.3. UCB-MC Isolation and Characterization

All UCB-MC units were collected from healthy donors with a gestation period of 37–40 weeks in maternity public hospitals in Kazan. Blood collections were carried out into single blood bags of 250 mL, with the blood preservative CPDA-1 (GCMS, Republic of Korea). Exclusion criteria were maternal infections or viral diseases. Isolations of mononuclear cells were conducted within 16–18 h after blood collection. Nucleated blood cells were isolated using SepMate ™-50 tubes according to the manufacturer’s protocol (STEMCELL Technologies Inc., Vancouver, BC, Canada). The viability of the isolated cells was determined in a hemocytometer with a 0.4% trypan blue solution. Cell viability, as measured by trypan blue exclusion, was >97%. The immune phenotype of isolated cells was analyzed by staining with monoclonal antibodies CD45—PerCP (BioLegend, San Diego, CA, USA), CD3-FITC (BioLegend, San Diego, CA, USA) CD14-APC/Cy7 (BioLegend, San Diego, CA, USA), CD38-APC/Cy7 (BioLegend, San Diego, CA, USA) CD34-FITC (BioLegend, San Diego, CA, USA), CD90-PE/Cy5 (BioLegend, San Diego, CA, USA). Expression of CD markers were analyzed by flow cytometry using BD FACS Aria III (BD bioscience, San Jose, CA, USA)

### 4.4. Analysis of Adenoviral Transduction of hUCB-MCs

Genetic modification of hUCB-MCs with recombinant adenoviruses (MOI 10 for each virus) was performed according to a previously developed protocol [77].

The efficiency of genetic modification was assessed after 72 h by means of fluorescent microscopy on AxioObserverZ1 (Carl Zeiss, Oberkochen, Germany) and flow cytometry using BD FACS Aria III (BD Bioscience, San Jose, CA, USA).

### 4.5. Total RNA Extraction and RT-qPCR

Analysis of the mRNA expression of VEGF165, FGF2, and SDF1α in genetically modified cells and isolated Matrigel implants was carried out by qPCR with further statistical analysis. Isolation of total RNA was performed by using the TRIzol (Thermo Fisher Scientific, Waltham, MA, USA) reagent according to the manufacturer’s recommendations with further cDNA synthesis. Real-Time PCR was set up on the Real-Time CFX96 Touch (BioRad Laboratories, CA, USA). The nucleotide sequences of the primers and probes used in RT-qPCR are mentioned in Table 2. All reactions for each sample were performed in triplicate with a further calculation of ΔΔCt values and normalization to the housekeeping gene of β-actin rRNA. Standard curves for quantitative analysis were created using serial dilutions of plasmid DNA with corresponding inserts (VEGF, FGF2, and SDF1α). Expression of target genes in non-transduced UCB-MCs was considered 100%. The level of the murine target gene mCD31 was normalized to the mouse housekeeping gene of mGAPDH. The statistical analysis of the obtained results was carried out in MS Excel 2007 with further calculation using U criteria (Mann-Whitney).

### 4.6. Analysis of Cytokines and Chemokines

Supernatant cytokine profiles were analyzed using Bio-Plex Pro Human Cytokine 27-plex Panel and Bio-Plex Human Cytokine 21-plex Panel (Bio-Rad, Hercules, CA, USA) multiplex magnetic bead-based antibody detection kits, following the manufacturer’s instructions. Supernatant aliquots (50 µL) were used for analysis, with a minimum of 50 beads per analyte acquired. Median fluorescence intensities were measured using a Bioplex 200 (Bio-Rad, Hercules, CA, USA) analyzer. Data collected were analyzed with MasterPlex CT control software and MasterPlex QT analysis software (Hitachi Software, San Bruno, CA, USA). Standard curves for each analyte were generated using standards provided by the manufacturer.

### 4.7. In Vivo Experiments

In vivo experiments were performed using immune-deficient mice of Balb/c nude lineage of both sexes for 7–8 weeks. The animals were bred using the animal facilities in Puschino’s laboratory. After quarantine, animals were held in an SPF vivarium with HEPA filters according to GLP standards. In the area of the withers, mice were subcutaneously injected with 2 million human transduced or native UCB-MCs mixed with 300 µL of Matrigel matrix. Female and male Balb/c nude mice were randomly assigned to a few groups: 1. Matrigel without UCB-MCs; 2. UCB-MCs transduced with Ad-EGFP in Matrigel; 3. UCB-MCs transduced with Ad-VEGF165 in Matrigel; 4. UCB-MCs transduced with a combination of adenoviruses Ad-VEGF165; Ad-SDF1α and Ad-FGF2 in Matrigel. All experiments were performed in quadruplicates. After seven days post-transplantation mice were taken from the experiment. The status of subcutaneous Matrigel implants was evaluated visually, and concentrations of hemoglobin were evaluated. Levels of the expression of therapeutic genes were analyzed by RT-qPCR. Production of therapeutic proteins was assessed via immunohistochemistry.

### 4.8. Analysis of Hemoglobin Concentration

The analysis of hemoglobin concentration in subcutaneous implants was evaluated colorimetrically. Implants were balanced by weight and homogenized in DPBS using the Mini-Bead Beater-16 (BioSpec, Bartlesville, OK, USA) with zirconium beads (d = 2 mm for 100 mg), during 2 cycles for 20 sec each. The obtained homogenates were centrifuged at 15,000× *g* for 15 min. Supernatants containing hemoglobin were examined on a microplate reader Tecan Infinite Pro 2000 with an OD of 540 nm (Tecan Austria GmbH, Grödig, Austria).

### 4.9. Histological Analysis

For histological analysis, Matrigel implants were fixed in a 10% buffered formalin solution for 48 h. After fixation, implants were dehydrated in increasing concentrations of ethanol and embedded in paraffin (Histomix, Biovitrum, Saint Petersburg, Russia). Paraffin slides with 5 µm thickness were prepared at the rotary microtome HM 355S (Thermo Fisher Scientific, Waltham, MA, USA). For general morphological characterization, slides were deparaffinized and stained with hematoxylin and eosin according to the standard protocol. For immunological studies, serial sections were deparaffinized and incubated in a citric buffer for 30 min to unmask epitopes. Cell membranes were permeabilized in a 0.1% solution of Tween-20 in PBS. Non-specific binding was blocked by incubation in a 10% solution of donkey serum for 30 min. Sections were stained with the antibodies to VEGF (mab293), FGF2 (sc-1390), and SFD1α (sc-28876), diluted 1:100 for 1 h at room temperature. After washing sections were stained for 1 h with secondary antibodies at room temperature followed by washing and DAPI staining (1:50,000 dilution) for 10 min. Primary and secondary antibodies are shown in Table 3. The microvessel density (MVD) was examined by counting the vessels in each implant, reported as the vessel number per square millimeter (vessel/mm^2^). Visualization of the results was performed on a scanning laser microscope LSM780 (Carl Zeiss, Oberkochen, Germany). Image analysis was performed using Image J software (https://fiji.sc/ (accessed on 18 December 2021)).

### 4.10. Statistical Analysis

GraphPad Prism^®^ 7 software was used to show all data reports (GraphPad, Inc., La Jolla, CA, USA). The data are presented as the mean ± standard error (SE). *p*-values were analyzed using a one-way analysis of variance (ANOVA) followed by Tukey’s test. Statistical significance is denoted by *p* < 0.05. All tests with animals, morphometric and statistical analyses were performed in a “blinded” manner with respect to the experimental groups.

## 5. Conclusions

The study suggests that UCB-MCs continue to be an essential source of stem cells for therapy and various gene-cell strategies for tissue regeneration. It is important to emphasize that the transplantation of genetically modified UCB-MCs is safer and more effective than direct gene therapy. Human UCB-MCs can be efficiently simultaneously modified with adenoviral vectors encoding VEGF, FGF2, and SDF1α. Modified UCB-MCs overexpress recombinant genes. Genetic modification of cells with recombinant adenoviruses (MOI 10) does not affect the profile of secreted pro- and anti-inflammatory cytokines, chemokines, or growth factors, except for an increase in the synthesis of recombinant proteins by cells. Modified cells can induce the formation of new vessels. Although many unresolved problems remain before modified UCB-MCs can be applied in the clinic, our results remain promising in terms of the induction of therapeutic angiogenesis in future clinical trials, including the treatment of decompensated forms.

## Figures and Tables

**Figure 1 ijms-24-04396-f001:**
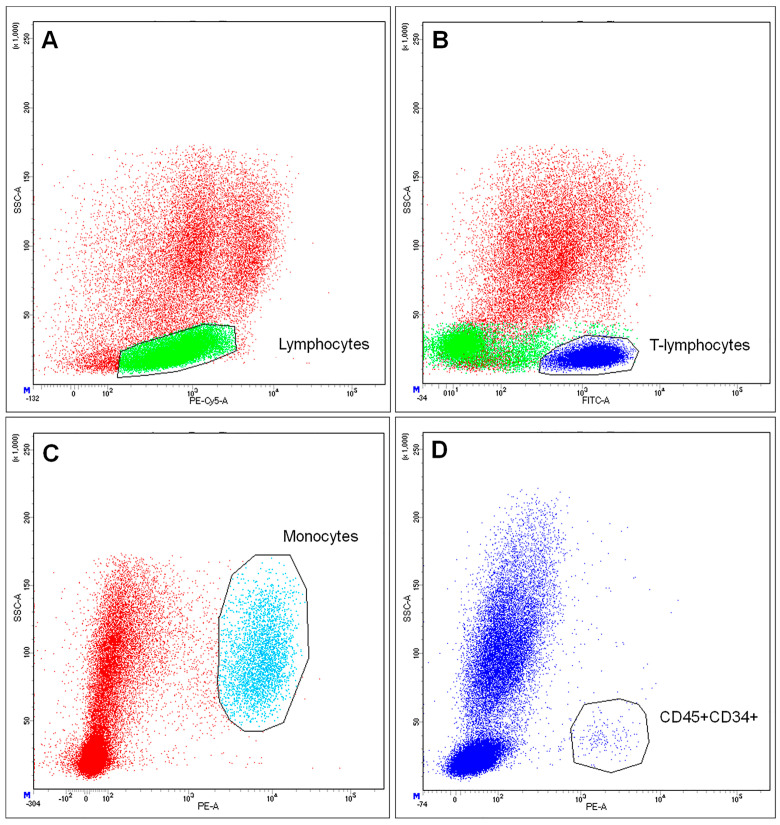
Determination of immune phenotype of umbilical cord blood mononuclear cells. (**A**) staining for CD45, selected green area–Lymphocytes; (**B**) staining for CD3, selected dark blue area–T-lymphocytes; (**C**) staining for CD14, selected blue area–Monocytes; (**D**) staining for CD34, selected events–CD45+ CD34+ cells.

**Figure 2 ijms-24-04396-f002:**
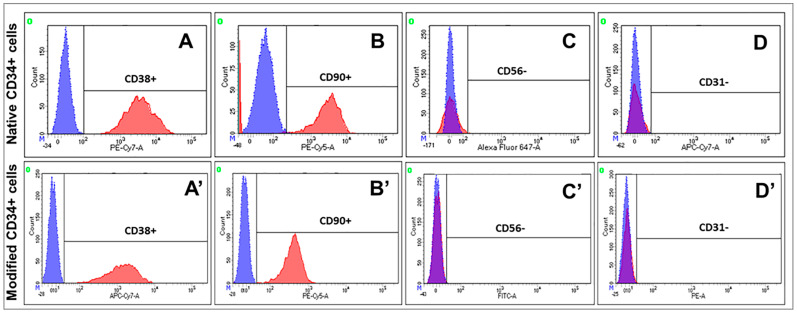
Determination of immune phenotype of CD34 positive cells. (**A**,**A’**) staining for CD38+; (**B**,**B’**) staining for CD90+; (**C**,**C’**) staining for CD56 (**D**,**D’**) staining for CD31. Native CD34+ cells—non-treated cells (UCB-MC-NTC). Modified CD34+ cells—cells modified with Ad-VEGF165, Ad-FGF2, and Ad-SDF1α (UCB-MC-VEGF-FGF2-SDF1α). Red peaks—staining with antibodies; blue peaks— negative control (isotype control).

**Figure 3 ijms-24-04396-f003:**
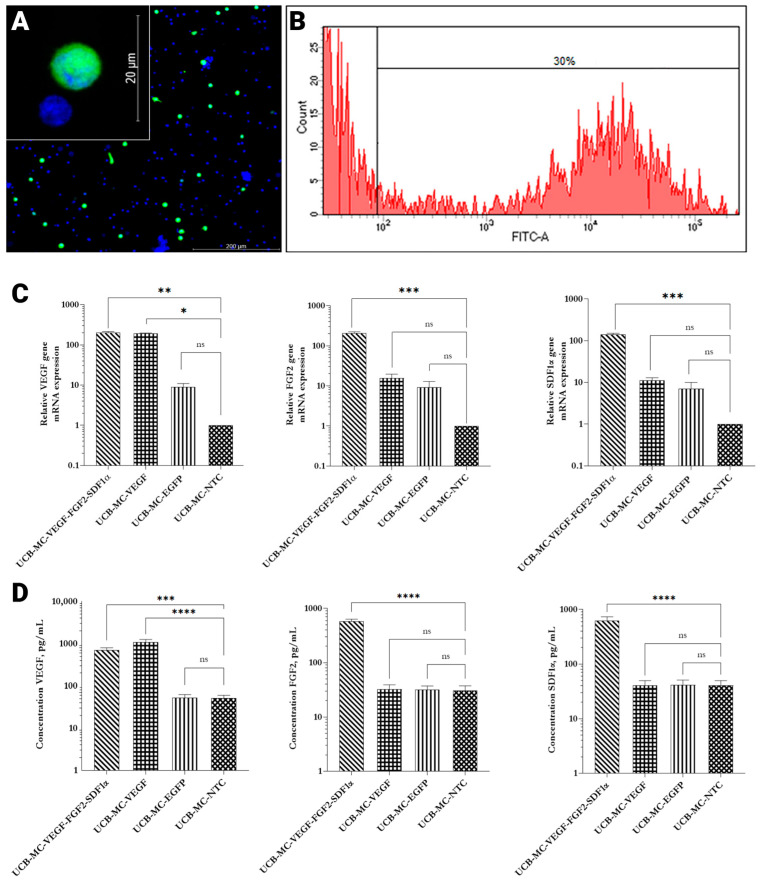
Analysis of transgenes expression in genetically modified UCB-MCs in vitro, 72 h post-transduction (MOI 10). (**A**) Fluorescent microscopy of EGFP+ cells (green fluorescence). Nuclei were stained with Hoechst 33342 (blue fluorescence). Scale bar 200 µm, 20 µm (inset). (**B**) Flow cytometry analysis of Ad-EGFP genetically modified UCB-MCs. Approximately 30% of genetically modified UCB-MCs exhibit green fluorescence. (**C**) Relative expression mRNA of therapeutic genes in genetically modified hUCB-MCs (*n* = 3). (**D**) The concentration of secreted proteins VEGF, FGF, and SDF1α in supernatants collected from genetically modified umbilical cord blood mononuclear cells (*n* = 8). UCB-MC-VEGF-FGF2-SDF1α—umbilical cord blood mononuclear cells modified with Ad-VEGF165, Ad-FGF2, and Ad-SDF1α. UCB-MC-VEGF—umbilical cord blood mononuclear cells modified with AdVEGF165. UCB-MC-EGFP—umbilical cord blood mononuclear cells modified with Ad-EGFP. UCB-MC-NTC—non-treated cells. Data presented as average ± standard error (SE); *p* < 0.05 * statistically significant differences compared to control (** *p* < 0.01; *** *p* < 0.001; **** *p* < 0.0001 ns—non-significant).

**Figure 4 ijms-24-04396-f004:**
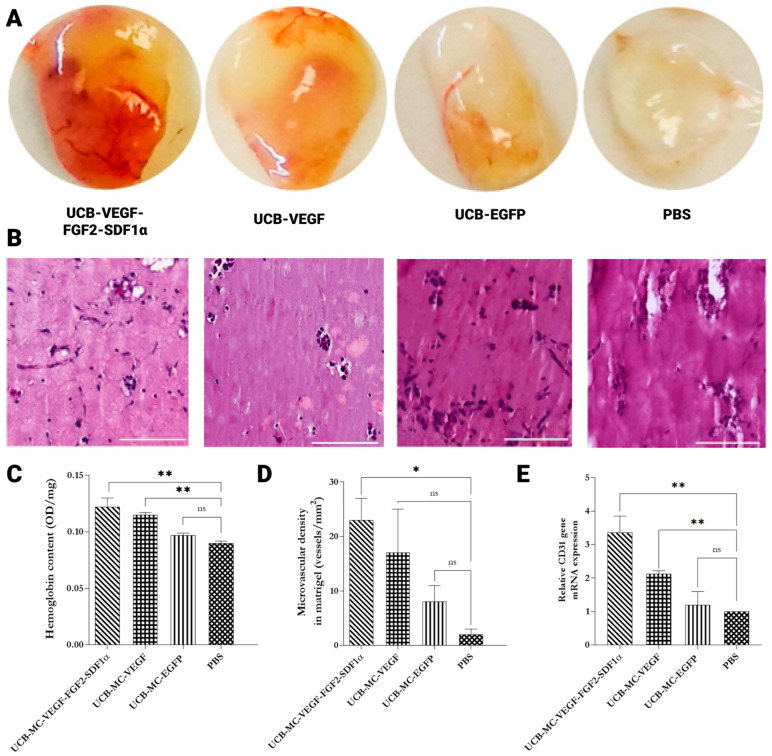
UCB-MC engineered with recombinant adenoviruses promotes angiogenesis in an in vivo Matrigel plug assay. Matrigel plugs collected at one-week post-implantation were appraised. (**A**) Representative gross morphological assessment and (**B**) microphotographs of harvested implants’ hematoxylin & eosin (H&E) staining. Scale bar 100 μm. (**C**) Quantification of hemoglobin concentration in each Matrigel plug (*n* = 4). (**D**) Histomorphometric analysis of vascularization process, number of blood vessels in frozen sections of the Matrigel plugs. (**E**) RT-qPCR quantification of CD31 mRNA in Matrigel plugs. UCB-MC-VEGF-FGF2- SDF1α—umbilical cord blood mononuclear cells modified with Ad-VEGF165, Ad-FGF2, and Ad-SDF1α. UCB-MC-VEGF—umbilical cord blood mononuclear cells modified with Ad-VEGF165. UCB-MC-EGFP—umbilical cord blood mononuclear cells modified with Ad-EGFP. PBS—Matrigel containing PBS. Data presented as average ± standard error (SE); *p* < 0.05 * statistically significant differences compared to control (*n* = 3; ** *p* < 0.01; ns—non-significant).

**Figure 5 ijms-24-04396-f005:**
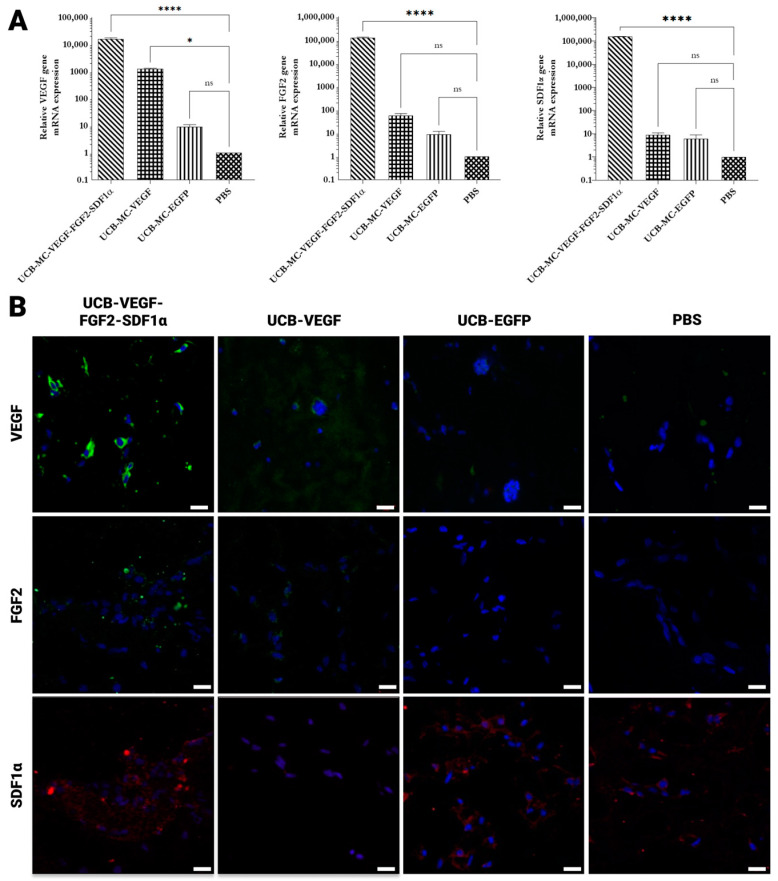
Analysis of transgenes expression in isolated Matrigel plugs one-week post-implantation. (**A**) Increasing concentrations of VEGF, FGF2, and SDF1α mRNA levels in Matrigel plugs. RT-qPCR analysis was performed on the total RNA extracted from gelled Matrigel samples. (**B**) Immunohistochemical analysis of Matrigel plugs. Staining for VEGF (green), FGF2 (green), and SDF1α (red). Nuclei were counterstained using a DAPI solution (4′,6-diamidino-2-phenylindole) (blue). Scale bars 50 µm. UCB-MC-VEGF-FGF2-SDF1—umbilical cord blood mononuclear cells modified with Ad-VEGF165, Ad-FGF2, and Ad-SDF1α. UCB-MC-VEGF—umbilical cord blood mononuclear cells modified with Ad-VEGF165. UCB-MC-EGFP—umbilical cord blood mononuclear cells modified with Ad-EGFP. PBS—Matrigel containing PBS. Data presented as average ± standard error (SE); *p* < 0.05 * statistically significant differences compared to control (*n* = 3; **** *p* < 0.0001; ns—non-significant.

**Table 1 ijms-24-04396-t001:** Profile of cytokines, chemokines, and growth factors secreted by genetically modified human umbilical cord blood mononuclear cells (pg/mL).

Analytes	UCB-MC- VEGF-FGF2-SDF1α	±SE	UCB-MC- VEGF	±SE	UCB-MC- EGFP	±SE	UCB-MC- NTC	±SE
**IL-1β**	69.74	40.13	99.93	75.35	80.73	51.62	105.13	71.47
**IL-1α**	69.70	45.97	63.22	46.65	54.39	41.85	66.03	49.29
**IL-2**	59.47	33.06	42.08	22.04	25.98	10.33	25.04	11.27
**IL-3**	32.79	27.69	19.13	10.13	13.95	7.79	27.93	20.94
**IL-4**	6.66	3.57	6.38	4.01	5.83	3.66	5.80	3.52
**IL-5**	100.61	55.99	95.73	57.98	108.39	70.69	125.89	79.67
**IL-6**	5034.98	1779.01	4337.05	1813.94	4357.59	1815.39	4697.32	1818.03
**IL-7**	4.16	1.35	3.93	1.38	4.07	1.37	3.60	1.40
**IL-8**	1,250,075.63	1,235,541.48	912,305.50	859,150.84	818,724.14	803,905.26	1,245,513.75	1,232,516.25
**IL-9**	27.19	11.69	28.90	11.99	26.39	12.43	27.37	13.48
**IL-10**	45.47	20.64	42.49	21.28	43.65	18.37	42.49	21.09
**IL-12p40**	206.89	53.84	178.88	51.16	181.16	61.56	204.02	69.32
**IL-12p70**	12.46	4.89	17.68	7.54	17.25	9.49	11.73	3.85
**IL-13**	2.29	0.58	2.28	0.57	2.37	0.76	2.38	0.71
**IL-15**	270.89	74.22	268.08	82.28	269.57	83.57	271.49	94.11
**IL-16**	422.01	102.13	413.50	91.00	383.75	81.51	396.14	85.94
**IL-17**	31.08	13.63	26.312	10.39	18.20	5.07	16.37	5.79
**IL-18**	9.14	3.96	10.08	5.06	6.32	2.31	5.85	1.94
**IL-1ra**	2126.49	773.43	1643.84	749.23	1545.21	692.37	1432.40	787.78
**IL-2Rα**	42.72	13.75	65.02	30.70	33.75	14.06	40.06	15.24
**G-CSF**	1270.38	721.68	1169.91	793.85	1264.01	866.29	1310.72	883.87
**M-CSF**	15.36	5.21	17.41	5.92	14.55	4.82	20.66	9.79
**GM-CSF**	30.27	16.49	30.14	14.82	23.83	11.51	25.07	10.86
**PDGF-bb**	262.31	79.05	242.27	31.67	224.96	62.40	193.86	44.27
**HGF**	155.12	63.49	182.51	76.23	185.92	75.70	214.67	101.40
**β-NGF**	5.33	1.93	5.98	3.32	3.15	1.33	4.24	1.50
**SCF**	104.75	47.36	94.22	60.91	77.45	47.19	93.06	47.29
**SCGF-β**	13,765.94	9382.70	9029.33	4187.62	7519.49	3573.11	9416.14	4526.52
**LIF**	83.69	32.05	69.63	32.17	63.90	29.94	65.40	32.74
**MIF**	6675.05	4487.06	6253.82	3726.91	4901.44	2670.02	6055.06	3408.90
**IFN-** **γ**	110.16	33.54	103.33	36.70	96.52	33.54	120.61	50.39
**IFN-α2**	21.65	5.38	23.15	6.72	17.06	5.53	18.21	6.05
**TNF-α**	706.57	339.62	636.71	343.12	686.63	409.36	612.62	369.89
**TNF-β**	65.06	20.03	45.37	14.01	85.01	30.49	56.53	18.19
**TRAIL**	309.92	160.98	260.55	168.85	263.23	164.56	266.27	165.56
**IP-10**	2946.86	2203.50	2756.89	2121.07	461.92	218.19	494.25	238.09
**MCP-1**	5878.67	1279.98	5633.98	1331.15	5927.77	1267.73	5548.28	1370.69
**MCP-3**	759.58	650.42	752.86	651.29	748.83	652.17	786.16	648.42
**MIP-1α**	27,907.69	27,255.16	62,599.84	61,948.04	580.24	283.44	575.48	284.73
**MIP-1β**	2085.04	1401.47	2114.97	1399.79	2050.55	1407.29	1838.60	1411.79
**RANTES**	944.01	102.62	910.36	211.74	807.69	146.79	789.65	207.02
**Eotaxin**	10.89	3.84	8.24	2.94	7.34	1.93	7.35	1.95
**CTACK**	222.07	189.83	222.26	189.11	121.43	93.65	160.64	127.47
**GROα**	18,520.81	11,913.68	20,273.64	11,681.33	18,519.01	11,844.77	53,200.26	33,134.06
**MIG**	262.56	191.60	282.09	192.34	228.99	142.93	315.42	178.72
**VEGF**	701.94	96.99	1087.12	169.11	52.31	10.36	51.75	8.65
**FGF-2**	576.27	57.83	32.36	6.65	32.03	4.98	30.71	6.59
**SDF-1α**	622.39	113.07	40.59	9.05	41.07	9.66	40.41	9.10

**Table 2 ijms-24-04396-t002:** Nucleotide sequences of primers used for RT-qPCR.

Name	Nucleotide Sequence
β-actin-TM-F (human)	GCGAGAAGATGACCCAGGATC
β-actin-TM-R (human)	CCAGTGGTACGGCCAGAGG
β-actin-TMprobe (human)	CCAGCCATGTACGTTGCTATCCAGGC
hVEGF-TM49F	TACCTCCACCATGCCAAGTG
hVEGF-TM110R	TGATTCTGCCCTCCTCCTTCT
hVEGF-TMProbe	TCCCAGGCTGCACCCATGG
hFGF2-TM134F	CCGACGGCCGAGTTGAC
hFGF2-TM203R	TCTTCTGCTTGAAGTTGTAGCTTGA
hFGF2-TMprobe	CCGGGAGAAGAGCGACCCTCAC
hSDF1-TM-F	TGACCGCTAAAGTGGTCGTC
hSDF1-TM-R	ACGTGGCTCTCAAAGAACCT
hSDF1-TMprobe	CCCTGTGCCTGTCCGATGGA
mCD31-F	CGGTTATGATGATGTTTCTGGA
mCD31-R	AAGGGAGGACACTTCCACTTCT
mGAPDH-F	GAAGGTCGGTGTGAACGGATT
mGAPDH-R	TGACTGTGCCGTTGAATTTG

**Table 3 ijms-24-04396-t003:** Primary and secondary antibodies used in immunofluorescent staining.

Antibody	Host	Dilution	Source
SFD1-α (sc-28876)	Rabbit	1:100	Abcam
FGF2 (SC-1390)	Goat	1:100	Santa Cruz
VEGF (mab293)	Mouse	1:100	R&D Systems
Anti- Mouse IgG Alexa Fluor 488	Goat	1:200	Invitrogen
Anti-Rabbit IgG H&L Alexa Fluor 647	Goat	1:200	Invitrogen
Anti-Rabbit IgG H&L Alexa Fluor 488	Goat	1:200	Abcam

## Data Availability

The data presented in this study are available on request from the corresponding author.

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
