# Peer review of "Induction of Angiogenesis by Genetically Modified Human Umbilical Cord Blood Mononuclear Cells"

_ijms, 2023, doi:10.3390/ijms24054396_

Round 1

Reviewer 1 Report

In this study, the authors found that hUCB-MCs can be efficiently simultaneously modified with several adenoviral vectors. Modified UCB-MCs overexpress recombinant genes and proteins. hUCB-MCs genetically modified with therapeutic genes induce the formation of new vessels. This current study demonstrates the employment of gene-engineered UCB-MC as an advanced strategy for stimulating of angiogenesis and possible therapy of cardiovascular diseases and diabetic cardiopathy.

This paper could be accepted for publication.

Author Response

Thank you for your time and hard work

Reviewer 2 Report

In this manuscript, Gatina and colleagues attempted to decipher how genetically modified UCB-MCs could improve regenerative medicine technics by the stimulation of angiogenesis. To do so, they used adenovirus constructs to induce the expression of arbitrary selected growth factors in a mixed population of cells obtained from cord blood. The strategy is interesting but this work lacks controls in experimental design and lacks a clear hypothesis of mechanism. Indeed, in addition to not presenting the right controls, the angiogenic capacity of the genetically modified cells is only evaluated using the matrigel plug assay.  For this reason, the manuscript is not suitable for publication as it currently is. In order to strengthen the conclusion and reach enough significance, I believe that several new key and control experiments would be required, especially the ones recommended below.

Major concerns: 

-       Figure 1: after the determination of the phenotype of UCB-MC used, the authors should perform the same experiment in the genetically modified cells in order to understand if the genetic modification leads to the differentiation of cells. ECFC markers and endothelial cell markers should be looked at. This would really strengthen the manuscript as a proper characterization of the modified cells should be performed.

-       In general, The UCB-MC-VEGF-FGF2-SDF1alpha group don’t have a proper control group. At least SDF1 alone and FGF alone should be added to the design of the study?

-       In the same perspective, the authors should better explain the choice of groups. Why compare VEGF-FGF2-SDF1 cells to EGFP cells?

-       Table 3: 

o   The cytokine profile would require more donors giving the high variability

o   The authors discuss the adenovirus treatment effect with activation of secretion of cytokines by activation of the immune system. However, the control group is non treated. How the authors explain this discrepancy?

o   The Ad genetic modification induces a dramatic variability of the protein expression targeted by the genetic modification. How authors explain this? This should be discussed.

-       Figure 3: 

o   The controls previously suggested are missing here. As mentioned by authors, SDF1 is able to recruit EPCs or other cell types that can support angiogenesis. Consequently, to better understand the mechanisms, controls groups should be added to the design of the study

o   The Matrigel-plug experiment lack of an important control group that is strictly mandatory : non treated UCB-MC

o   Could the authors provide picture used for microvascular density? What ab was used? It would be interesting to use an anti-human and an anti-mouse antibodies in order to differentiate between the vasculogenesis occurring in the plug and the cell recruitment. 

o   Again, why EGFP compared to the other 2 groups? the scope is ambiguous here.

-       Figure 4: 

o   How authors observed the recruited fibroblasts in the plugs?

o   It looks like there are post-fixation issues in some of the pictures. Are the authors confident with the staining? 

o   Could the authors explain better the color code for the staining.

-       Other angiogenesis assays should be added to reinforce the manuscript. At least in vitro experiments that would also contribute to the characterization of the cells.

Minor concerns:

-       MM table 2: Is there a mistake in the ab list? The secondaries are all anti-rabbit?

-       Microvessel density analysis material and methods are missing

-       References about state-of-the-art methods to study angiogenesis should be added

Author Response

Major concerns: 

  1. Figure 1: after the determination of the phenotype of UCB-MC used, the authors should perform the same experiment in the genetically modified cells in order to understand if the genetic modification leads to the differentiation of cells. ECFC markers and endothelial cell markers should be looked at. This would really strengthen the manuscript as a proper characterization of the modified cells should be performed.

Response 1: Thank you very much for your comments.

The manuscript has been revised. In this study, we did not focus on the characterization of the isolated cells. In the study, we showed the genetically modified cells had the same immunophenotype as the unmodified cells. The study showed that the genetically modified cells had the same immunophenotype as the unmodified cells. Moreover, in previously published research, we have shown that genetic modification does not affect the mRNA landscape of the cells. Furthermore, we have shown in previous work that genetic modifications do not alter the reorganization of translated mRNAs (transcriptome) in transduced cells. We also did not find any differences in the mRNA expression profile of ECFC CD-markers after cell modification with recombinant adenoviruses (https://doi.org/10.1182/blood-2020-138664).

  1. In general, The UCB-MC-VEGF-FGF2-SDF1α group don’t have a proper control group. At least SDF1 alone and FGF alone should be added to the design of the study?

-In the same perspective, the authors should better explain the choice of groups. Why compare VEGF-FGF2-SDF1 cells to EGFP cells?

Response 2: Thank you very much for your comment.

In our in vitro studies, the controls for UCB-MC-VEGF-FGF2-SDF1α are either Ad-EGFP-modified cells or unmodified cells. Our goal was not to analyze each type of modification in detail. In the in vivo block for UCB-MC-VEGF-FGF2-SDF1α, the controls were Ad-EGFP-modified cells, as well as fragments of the matrix without cells (PBS). Comparing VEGF-FGF2-SDF1 cells to EGFP cells, so we evaluate the effect of genetic modification.

Table 3.

  1. The cytokine profile would require more donors giving the high variability

Response 3:  You may be right; at the same time, the number of donors is adequate in the context of the study design.  In this study, isolated and characterized cells were used in subsequent stages of the research. Therefore, the viewed therapeutic effect was determined by these cells. Here, the biological/therapeutic effect is determined by the secretion of recombinant proteins by cells. As well, these studies were previously conducted by us, which showed changes in the expression of only recombinant proteins.

  1. The authors discuss the adenovirus treatment effect with activation of secretion of cytokines by activation of the immune system. However, the control group is non treated. How the authors explain this discrepancy?

Response 4:  In this case, the control group is UCB-MC-EGFP. We are evaluating the expression of therapeutic proteins and the effects of genetic modification on the secreted factor profile. Additionally, we are considering the natural secretome profile of the cells in our analysis. Our experiments have shown no immune response activation, such as the synthesis of cytokines and chemokines, due to genetic modification.

  1. The Ad genetic modification induces a dramatic variability of the protein expression targeted by the genetic modification. How authors explain this? This should be discussed.

Response 5: In this case, this is the expected result. We are trying to achieve this effect through genetic modification. The modification UCB-MCs with Ads leads to the overexpression of recombinant proteins (growth factors -  VEGF, FGF2, SDF1). At the same time, the profile of the other investigated factors is not affected.

Figure 3.

  1. The controls previously suggested are missing here. As mentioned by authors, SDF1 is able to recruit EPCs or other cell types that can support angiogenesis. Consequently, to better understand the mechanisms, controls groups should be added to the design of the study

Response 6:  We agree with your comments and have made the necessary changes. In this study, the therapeutic agent is UCB-MC-VEGF-FGF2-SDF1α, so that we will focus on this group. The therapeutic strategy implemented involves using UCB-MC modified with three Ads (Ad-VEGF, Ad-FGF2, Ad-SDF1α). The control groups in this block are UCB-MC-EGFP – umbilical cord blood mononuclear cells modified with Ad-EGFP and PBS – Matrigel containing PBS. Of course, the consideration of groups, Ad-FGF2, Ad-SDF1α and their combinations is obvious. However, studies of this plan were carried out by other research groups and us earlier.

  1. The Matrigel-plug experiment lack of an important control group that is strictly mandatory: non treated UCB-MC

Response 7: In this case, the untreated UCB-MC group is represented by the UCB-MC-EGFP group. Moreover, this group characterizes the absence of the effect of genetic modification on UCB-MC. In particular, we previously showed that the genetic modification of cells (Ad-EGFP) does not affect the profile of secreted factors (https://doi.org/10.1182/blood-2020-138664, https://doi.org/10.3390/ijms21186858). Therefore, this control group is sufficient for our study.

 We appreciate your feedback and will consider adding additional control groups in future studies.

  1. Could the authors provide picture used for microvascular density? What ab was used? It would be interesting to use an anti-human and an anti-mouse antibodies in order to differentiate between the vasculogenesis occurring in the plug and the cell recruitment.

Response 8: The use of specific anti-human and anti-mouse antibodies certainly has a potential scientific interest. Visualization of microvascular density was performed without the use of antibodies. The assessment of the formed vessels was carried out visually with hematoxylin & eosin (H&E) staining. However, it is also worth considering that the formed vessels are derived from the mouse component. Transplanted cells secrete growth and trophic factors (paracrine stimulation) and recruit recipient cells.

  1. Again, why EGFP compared to the other 2 groups? the scope is ambiguous here.

Response 9:

In this case, using EGFP as a control group is appropriate for our study. In the study, we assess the impact of genetic modification. Native cells (in our case, AD-EGFP-modified cells, which do not differ from unmodified (NTC) cells) can exercise a pro-angiogenic effect by secreting various proactive molecules.

Figure 4.

  1.  How authors observed the recruited fibroblasts in the plugs?

Response 10: Recruitment of cells was assessed by imaging histological sections and by counting migrating fibroblasts

  1.  It looks like there are post-fixation issues in some of the pictures. Are the authors confident with the staining?

Response 11: Thank you for your comment. The authors are confident in the results presented. Additionally, histological data is consistent with the results of the RT-qPCR analysis.

  1.  Could the authors explain better the color code for the staining.

Response 12: Necessary changes made.

  1.  Other angiogenesis assays should be added to reinforce the manuscript. At least in vitro experiments that would also contribute to the characterization of the cells.

Response 13: Thank you very much for your comment. In previous research, we conducted whole transcriptome profiling and demonstrated the impact of supernatants collected from genetically modified cells on HUVEC (in vitro) proliferation. We have also shown the ability of modified UCB to differentiate into different cell types and stimulate angiogenesis in various in vivo models (https://doi.org/10.3390/ijms21186858, https://doi.org/10.1007/s10517-013-1999-2 , https://doi.org/10.1007/s11055-013-9778-0, https://doi.org/10.1007/s12035-016-0017-x, https://doi.org/10.1038/sc.2015.162, https://doi.org/10.3390/cells10020432).

Minor concerns:

  1.   MM table 2: Is there a mistake in the ab list? The secondaries are all anti-rabbit?

Response 14: Thank you for your comment.  The necessary changes have been made.

  1.   Microvessel density analysis material and methods are missing

Response 15: Thank you for your comment. The necessary changes have been made.

  1. References about state-of-the-art methods to study angiogenesis should be added

Response 16: Thank you very much for your comment. Many methods are currently available to study angiogenesis (https://doi.org/10.3390/biomedicines7020037, https://doi.org/10.1155/2020/8857428). In the current article, we focused on one is the most classical methods for angiogenesis analysis.

Reviewer 3 Report

The manuscript entitled, ‘Induction of angiogenesis by genetically modified human umbilical cord blood mononuclear cells’ is very interesting and demanding in the field of angiogenesis to treat diseases associated with microvascular rarefaction. Although this study is possessing novel insights as well as the authors employed several in vitro and in vivo approaches, several issues need to be addressed before its publication.

1.      The whole manuscript is poorly written, as there are numerous mistakes with grammar and typos.

2.      On page 3, lines 96-97: Rewrite the sentence as ‘The HEK293A cells were infected with a coarse viral runoff to prepare necessary’.

3.      amounts of Ad-VEGF, Ad-FGF2, Ad-SDF1α, and Ad-EGFP adenoviruses.

4.      The abstract is missing the hypothesis of the study. The authors should provide a hypothesis for their study.

5.      On page 5, line 192: It will be, ‘the data are presented as the mean ± standard error’ as in figures 2, 3, and 4, the authors used mean ± standard error.

6.      On page 5, line 198: The heading, ‘Isolation and genetic modification of human UCB-MCs’ is not appropriate for a result section. Try to replace it with a sentence that conveys the major finding/s of the section under the heading, e.g., Isolated human UCB-MCs demonstrated increased viability.

7.      On page 5, lines 199-203: The following sentences, ‘Recombinant Ad-SDF1α adenovirus.……..the ficoll density’ should be under the material as and methods section.

8.      On page 6, lines 208-210: The sentence, ‘According to the obtained data CD34+ cells constituted 0,4% 208 of CD45+ cells, while 91.8% of these cells demonstrated CD45+CD34+CD38+ phenotype and 90% of these cells expressed CD45+CD34+CD90+ surface markers’ is not clear to me. Please rewrite the sentence.

9.      On page 6, line 211: The quality of figure 1 is not good; try to improve it. Additionally, try to rewrite the figure 1 legend as follows: (A) staining for CD45+; (B) staining for CD3+; (C) staining for CD14+; (D) staining for CD34+; (E)staining for CD38+; (F) staining for CD90+.

10.   On page 6, line 212: The heading, ‘Analysis of expression of transgenes in modified UCB-MCs in vitro’ is not appropriate for a result section. Try to replace it with a sentence that conveys the major finding/s of the section under the heading, e.g., Genetic modification of the UCB-MCs with recombinant adenoviruses was confirmed by EGFP expression in UCB-MCs.

11.   On page 7, line 227: The heading, ‘Effect of genetic modification of hUCB-MCs on secretion profile’ is not appropriate for a result section. Try to replace it with a sentence that conveys the major finding/s of the section under the heading, e.g., ‘genetically modified hUCB-MCs produce a broad range of cytokines, chemokines and growth factors’.

12.   On page 7, lines 228-232: The sentences, ‘hUCB-MCs were…… postinfection by Luminex assays’ should be moved to the Materials and methods section.

13.   On page 9: Table 3: it will be human umbilical cord blood mononuclear cells.

14.   On page 9: Below table 3, try to rewrite the sentences as they are confusing, ‘VSF – umbilical cord blood mononuclear cells modified with Ad-VEGF165, Ad-FGF2 and Ad-SDF1α. VEGF –umbilical cord blood mononuclear cells modified with Ad-VEGF165. EGFP – umbilical cord blood mononuclear cells modified with Ad-EGFP. NTC – non treated cells. ±SD – standard deviation, NTC – not transduction control’.

15.   On page 10: Figure 3 legend, panel (E) correct the sentence to RT-qPCR quantification of CD31 mRNA in Matrigel plugs.

16.    On page 12: Figure 4 legend, panel (A) Rewrite the sentence to ‘increasing concentrations of VEGF, FGF2, and SDF1α mRNA levels in Matrigel plugs’.

Author Response

  1. The whole manuscript is poorly written, as there are numerous mistakes with grammar and typos.

Response 1: Thank you very much for your comment. A revised and rewritten version of the manuscript has been submitted.

  1. On page 3, lines 96-97: Rewrite the sentence as ‘The HEK293A cells were infected with a coarse viral runoff to prepare necessary’ amounts of Ad-VEGF, Ad-FGF2, Ad-SDF1α, and Ad-EGFP adenoviruses.

Response 2: Thanks for your comments. Changes have been implemented in the manuscript.

  1. The abstract is missing the hypothesis of the study. The authors should provide a hypothesis for their study.

Response 3: The manuscript has been amended. The hypothesis has been formulated.

  1. On page 5, line 192: It will be, ‘the data are presented as the mean ± standard error’ as in figures 2, 3, and 4, the authors used mean ± standard error.

Response 4: Thanks for your comments. Changes have been implemented in the manuscript.

  1. On page 5, line 198: The heading, ‘Isolation and genetic modification of human UCB-MCs’ is not appropriate for a result section. Try to replace it with a sentence that conveys the major finding/s of the section under the heading, e.g., Isolated human UCB-MCs demonstrated increased viability.

Response 5: Thanks for your recommendation. Changes have been implemented in the manuscript.

  1. On page 5, lines 199-203: The following sentences, ‘Recombinant Ad-SDF1α adenovirus.……..the ficoll density’ should be under the material as and methods section.

Response 6: Thanks for your comments. Changes have been implemented in the manuscript.

  1. On page 6, lines 208-210: The sentence, ‘According to the obtained data CD34+ cells constituted 0,4% 208 of CD45+ cells, while 91.8% of these cells demonstrated CD45+CD34+CD38+ phenotype and 90% of these cells expressed CD45+CD34+CD90+ surface markers’ is not clear to me. Please rewrite the sentence.

Response 7: Thanks for your comments. Changes have been implemented in the manuscript.

  1. On page 6, line 211: The quality of figure 1 is not good; try to improve it. Additionally, try to rewrite the figure 1 legend as follows: (A) staining for CD45+; (B) staining for CD3+; (C) staining for CD14+; (D) staining for CD34+; (E)staining for CD38+; (F) staining for CD90+.

Response 8: Thanks for your comments. Changes have been implemented in the manuscript.

  1. On page 6, line 212: The heading, ‘Analysis of expression of transgenes in modified UCB-MCs in vitro’ is not appropriate for a result section. Try to replace it with a sentence that conveys the major finding/s of the section under the heading, e.g., Genetic modification of the UCB-MCs with recombinant adenoviruses was confirmed by EGFP expression in UCB-MCs.

Response 9: Thanks for your comments. Changes have been implemented in the manuscript.

  1. On page 7, line 227: The heading, ‘Effect of genetic modification of hUCB-MCs on secretion profile’ is not appropriate for a result section. Try to replace it with a sentence that conveys the major finding/s of the section under the heading, e.g., ‘genetically modified hUCB-MCs produce a broad range of cytokines, chemokines and growth factors’

Response 10: Thanks for your comments. Changes have been implemented in the manuscript.

  1. On page 7, lines 228-232: The sentences, ‘hUCB-MCs were…… postinfection by Luminex assays’ should be moved to the Materials and methods section.

Response 11: Thanks for your comments. Changes have been implemented in the manuscript.

  1. On page 9: Table 3: it will be human umbilical cord blood mononuclear cells.

Response 12: Thanks for your comments. Changes have been implemented in the manuscript.

  1. Below table 3, try to rewrite the sentences as they are confusing, ‘VSF – umbilical cord blood mononuclear cells modified with Ad-VEGF165, Ad-FGF2 and Ad-SDF1α. VEGF –umbilical cord blood mononuclear cells modified with Ad-VEGF165. EGFP – umbilical cord blood mononuclear cells modified with Ad-EGFP. NTC – non treated cells. ±SD – standard deviation, NTC – not transduction control’.

Response 13: Thanks for your comments. Changes have been implemented in the manuscript.

  1. On page 10: Figure 3 legend, panel (E) correct the sentence to RT-qPCR quantification of CD31 mRNA in Matrigel plugs.

Response 14: Thanks for your comments. Changes have been implemented in the manuscript.

15. On page 12: Figure 4 legend, panel (A) Rewrite the sentence to ‘increasing concentrations of VEGF, FGF2, and SDF1α mRNA levels in Matrigel plugs’.

Response 15: Thanks for your comments. Changes have been implemented in the manuscript.

Round 2

Round 1

In this manuscript, Gatina and colleagues attempted to decipher how genetically modified UCB-MCs could improve regenerative medicine technics by the stimulation of angiogenesis. To do so, they used adenovirus constructs to induce the expression of arbitrary selected growth factors in a mixed population of cells obtained from cord blood. The strategy is interesting but this work lacks controls in experimental design and lacks a clear hypothesis of mechanism. Indeed, in addition to not presenting the right controls, the angiogenic capacity of the genetically modified cells is only evaluated using the matrigel plug assay.  For this reason, the manuscript is not suitable for publication as it currently is. In order to strengthen the conclusion and reach enough significance, I believe that several new key and control experiments would be required, especially the ones recommended below.

Major concerns: 

-       Figure 1: after the determination of the phenotype of UCB-MC used, the authors should perform the same experiment in the genetically modified cells in order to understand if the genetic modification leads to the differentiation of cells. ECFC markers and endothelial cell markers should be looked at. This would really strengthen the manuscript as a proper characterization of the modified cells should be performed.

-       In general, The UCB-MC-VEGF-FGF2-SDF1alpha group don’t have a proper control group. At least SDF1 alone and FGF alone should be added to the design of the study?

-       In the same perspective, the authors should better explain the choice of groups. Why compare VEGF-FGF2-SDF1 cells to EGFP cells?

-       Table 3: 

o   The cytokine profile would require more donors giving the high variability

o   The authors discuss the adenovirus treatment effect with activation of secretion of cytokines by activation of the immune system. However, the control group is non treated. How the authors explain this discrepancy?

o   The Ad genetic modification induces a dramatic variability of the protein expression targeted by the genetic modification. How authors explain this? This should be discussed.

-       Figure 3: 

o   The controls previously suggested are missing here. As mentioned by authors, SDF1 is able to recruit EPCs or other cell types that can support angiogenesis. Consequently, to better understand the mechanisms, controls groups should be added to the design of the study

o   The Matrigel-plug experiment lack of an important control group that is strictly mandatory : non treated UCB-MC

o   Could the authors provide picture used for microvascular density? What ab was used? It would be interesting to use an anti-human and an anti-mouse antibodies in order to differentiate between the vasculogenesis occurring in the plug and the cell recruitment. 

o   Again, why EGFP compared to the other 2 groups? the scope is ambiguous here.

-       Figure 4: 

o   How authors observed the recruited fibroblasts in the plugs?

o   It looks like there are post-fixation issues in some of the pictures. Are the authors confident with the staining? 

o   Could the authors explain better the color code for the staining.

-       Other angiogenesis assays should be added to reinforce the manuscript. At least in vitro experiments that would also contribute to the characterization of the cells.

Minor concerns:

-       MM table 2: Is there a mistake in the ab list? The secondaries are all anti-rabbit?

-       Microvessel density analysis material and methods are missing

-       References about state-of-the-art methods to study angiogenesis should be added

Round 2

Comments in the Author's Notes in purple

Major concerns: 

  1. Figure 1: after the determination of the phenotype of UCB-MC used, the authors should perform the same experiment in the genetically modified cells in order to understand if the genetic modification leads to the differentiation of cells. ECFC markers and endothelial cell markers should be looked at. This would really strengthen the manuscript as a proper characterization of the modified cells should be performed.

Response 1: Thank you very much for your comments.

The manuscript has been revised. In this study, we did not focus on the characterization of the isolated cells. In the study, we showed the genetically modified cells had the same immunophenotype as the unmodified cells. The study showed that the genetically modified cells had the same immunophenotype as the unmodified cells. Moreover, in previously published research, we have shown that genetic modification does not affect the mRNA landscape of the cells. Furthermore, we have shown in previous work that genetic modifications do not alter the reorganization of translated mRNAs (transcriptome) in transduced cells. We also did not find any differences in the mRNA expression profile of ECFC CD-markers after cell modification with recombinant adenoviruses (https://doi.org/10.1182/blood-2020-138664).

I am sorry, the figure 1 seems to show the phenotype of UCB-MCs, which is already known. To give an input to this figure the authors need to add the same experiments but in the genetically modified cells. It means showing results of the experiments with figures and not only adding a sentence in the results section. If the authors 

 In general, The UCB-MC-VEGF-FGF2-SDF1α group don’t have a proper control group. At least SDF1 alone and FGF alone should be added to the design of the study?

-In the same perspective, the authors should better explain the choice of groups. Why compare VEGF-FGF2-SDF1 cells to EGFP cells?

Response 2: Thank you very much for your comment.

In our in vitro studies, the controls for UCB-MC-VEGF-FGF2-SDF1α are either Ad-EGFP-modified cells or unmodified cells. Our goal was not to analyze each type of modification in detail. In the in vivo block for UCB-MC-VEGF-FGF2-SDF1α, the controls were Ad-EGFP-modified cells, as well as fragments of the matrix without cells (PBS). Comparing VEGF-FGF2-SDF1 cells to EGFP cells, so we evaluate the effect of genetic modification.

Ad-EGFP is not a suitable control for VEGF-FGF2-SDF1α. Comparing VEGF-FGF2-SDF1 cells to EGFP cells don’t allow you to evaluate the effect of genetic modification, this is why proper controls are seriously lacking. In addition to adding the missing control groups, if the authors want to keep the EGFP comparison with VEGF-FGF2-SDF1α, these choices should be better discussed in the introduction and discussion. 

Table 3.

  1. The cytokine profile would require more donors giving the high variability

Response 3: You may be right; at the same time, the number of donors is adequate in the context of the study design. In this study, isolated and characterized cells were used in subsequent stages of the research. Therefore, the viewed therapeutic effect was determined by these cells. Here, the biological/therapeutic effect is determined by the secretion of recombinant proteins by cells. As well, these studies were previously conducted by us, which showed changes in the expression of only recombinant proteins.

Could the authors provide de statistical analysis? Given the values reported in the table and especially the SD values, we do not find any statistical differences. The extreme variability does not allow to conclude a priori. Even without statistics, how do you explain that the secretion of UCB-MC-VEGF is more than 3 times higher than for UCB-MC-VEGF-FGF2-SDF1α.

  1. The authors discuss the adenovirus treatment effect with activation of secretion of cytokines by activation of the immune system. However, the control group is non treated. How the authors explain this discrepancy?

Response 4: In this case, the control group is UCB-MC-EGFP. We are evaluating the expression of therapeutic proteins and the effects of genetic modification on the secreted factor profile. Additionally, we are considering the natural secretome profile of the cells in our analysis. Our experiments have shown no immune response activation, such as the synthesis of cytokines and chemokines, due to genetic modification.

UCB-MC-EGFP is not a proper control but OK for the point 4.

  1. The Ad genetic modification induces a dramatic variability of the protein expression targeted by the genetic modification. How authors explain this? This should be discussed.

Response 5: In this case, this is the expected result. We are trying to achieve this effect through genetic modification. The modification UCB-MCs with Ads leads to the overexpression of recombinant proteins (growth factors - VEGF, FGF2, SDF1). At the same time, the profile of the other investigated factors is not affected.

The problem is not inter-group variability but intra-group variability. SD values are sometimes at the same level or even greater than the mean. It seems difficult to discuss these results given the statistical flaws.

Figure 3.

  1. The controls previously suggested are missing here. As mentioned by authors, SDF1 is able to recruit EPCs or other cell types that can support angiogenesis. Consequently, to better understand the mechanisms, controls groups should be added to the design of the study

Response 6:  We agree with your comments and have made the necessary changes. In this study, the therapeutic agent is UCB-MC-VEGF-FGF2-SDF1α, so that we will focus on this group. The therapeutic strategy implemented involves using UCB-MC modified with three Ads (Ad-VEGF, Ad-FGF2, Ad-SDF1α). The control groups in this block are UCB-MC-EGFP – umbilical cord blood mononuclear cells modified with Ad-EGFP and PBS – Matrigel containing PBS. Of course, the consideration of groups, Ad-FGF2, Ad-SDF1α and their combinations is obvious. However, studies of this plan were carried out by other research groups and us earlier.

You need to show the advantage of genetically modified cells. In order to do this and show that UCB-MC-VEGF-FGF2-SDF1α is a therapeutic strategy that should be considered, you must use the right controls. Again, UCB-MC-EGFP is not a proper control. If such studies have been made, they should be introduced in the manuscript not only as a citation for material and methods. You must clarify the identification and choice of your therapeutic strategy.

  1. The Matrigel-plug experiment lack of an important control group that is strictly mandatory: non treated UCB-MC

Response 7: In this case, the untreated UCB-MC group is represented by the UCB-MC-EGFP group.Moreover, this group characterizes the absence of the effect of genetic modification on UCB-MC. In particular, we previously showed that the genetic modification of cells (Ad-EGFP) does not affect the profile of secreted factors (https://doi.org/10.1182/blood-2020-138664, https://doi.org/10.3390/ijms21186858). Therefore, this control group is sufficient for our study.

We appreciate your feedback and will consider adding additional control groups in future studies.

Again, UCB-MC-EGFP is not a proper control. If this paper will be published as it is. The authors need to clarify the choice of their control and pointing out the weakness of choosing such unappropriated control. The authors need to clarify the design of the study. When you designed the study you really chosen to have EGFP group as a control group? 

  1.  Could the authors provide picture used for microvascular density? What ab was used? It would be interesting to use an anti-human and an anti-mouse antibodies in order to differentiate between the vasculogenesis occurring in the plug and the cell recruitment.

Response 8: The use of specific anti-human and anti-mouse antibodies certainly has a potential scientific interest. Visualization of microvascular density was performed without the use of antibodies. The assessment of the formed vessels was carried out visually with hematoxylin & eosin (H&E) staining. However, it is also worth considering that the formed vessels are derived from the mouse component. Transplanted cells secrete growth and trophic factors (paracrine stimulation) and recruit recipient cells.

The authors could add another method to quantify the microvascular density. Vessels should be properly stained. HE pictures are not representative of the quantifications (VEGF and EGFP). These elements about cell recruitment and the origin of the endothelial cells giving raise to the vessel within the plug, should be at least discussed in the manuscript if any experiments are going to be run. 

  1. Again, why EGFP compared to the other 2 groups? the scope is ambiguous here.

Response 9: 

In this case, using EGFP as a control group is appropriate for our study. In the study, we assess the impact of genetic modification. Native cells (in our case, AD-EGFP-modified cells, which do not differ from unmodified (NTC) cells) can exercise a pro-angiogenic effect by secreting various proactive molecules.,

By using EGFP as a control you don’t assess the impact of genetic modification. You assess the difference between modifying EGFP or a cocktail of growth factors. EGFP modified cells are not native as obviously mentioned by their name. The authors should explain better the design of the study. Why the control group wasn’t added? 

Figure 4.

  1. How authors observed the recruited fibroblasts in the plugs?

Response 10: Recruitment of cells was assessed by imaging histological sections and by counting migrating fibroblasts

This is not answering the question? How were you able to identify the fibroblasts?

  1. It looks like there are post-fixation issues in some of the pictures. Are the authors confident with the staining?

Response 11: Thank you for your comment. The authors are confident in the results presented. Additionally, histological data is consistent with the results of the RT-qPCR analysis.

Could the authors provide other pictures? considering the pictures presented it means that there is high secretion of VEGF in the plugs injected with only PBS? 

  1. Could the authors explain better the color code for the staining.

Response 12: Necessary changes made.

OK

  1. Other angiogenesis assays should be added to reinforce the manuscript. At least in vitro experiments that would also contribute to the characterization of the cells.

Response 13: Thank you very much for your comment. In previous research, we conducted whole transcriptome profiling and demonstrated the impact of supernatants collected from genetically modified cells on HUVEC (in vitro) proliferation. We have also shown the ability of modified UCB to differentiate into different cell types and stimulate angiogenesis in various in vivo models (https://doi.org/10.3390/ijms21186858, https://doi.org/10.1007/s10517-013-1999-2 , https://doi.org/10.1007/s11055-013-9778-0, https://doi.org/10.1007/s12035-016-0017-x, https://doi.org/10.1038/sc.2015.162, https://doi.org/10.3390/cells10020432).

This should be added to the manuscript. However, none of these paper show the study of the genetic modifications presented in angiogenesis assays such as tubulogenesis assays, 3D sprouting, wound healing… 

Minor concerns:

  1. MM table 2: Is there a mistake in the ab list? The secondaries are all anti-rabbit?

Response 14: Thank you for your comment. The necessary changes have been made.

OK

  1. Microvessel density analysis material and methods are missing

Response 15: Thank you for your comment. The necessary changes have been made.

How vessels were identified? with presence of erythrocytes in the lumen? 

  1. References about state-of-the-art methods to study angiogenesis should be added

Response 16: Thank you very much for your comment. Many methods are currently available to study angiogenesis (https://doi.org/10.3390/biomedicines7020037, https://doi.org/10.1155/2020/8857428). In the current article, we focused on one is the most classical methods for angiogenesis analysis.

That's nice but I wasn't speaking for myself, it was a suggestion to add to your manuscript.

Round 2

Major concerns: 

  1. Figure 1: after the determination of the phenotype of UCB-MC used, the authors should perform the same experiment in the genetically modified cells in order to understand if the genetic modification leads to the differentiation of cells. ECFC markers and endothelial cell markers should be looked at. This would really strengthen the manuscript as a proper characterization of the modified cells should be performed.

Response 1: Thank you very much for your comments.

The manuscript has been revised. In this study, we did not focus on the characterization of the isolated cells. In the study, we showed the genetically modified cells had the same immunophenotype as the unmodified cells. The study showed that the genetically modified cells had the same immunophenotype as the unmodified cells. Moreover, in previously published research, we have shown that genetic modification does not affect the mRNA landscape of the cells. Furthermore, we have shown in previous work that genetic modifications do not alter the reorganization of translated mRNAs (transcriptome) in transduced cells. We also did not find any differences in the mRNA expression profile of ECFC CD-markers after cell modification with recombinant adenoviruses (https://doi.org/10.1182/blood-2020-138664).

I am sorry, the figure 1 seems to show the phenotype of UCB-MCs, which is already known. To give an input to this figure the authors need to add the same experiments but in the genetically modified cells. It means showing results of the experiments with figures and not only adding a sentence in the results section. If the authors 

Response: Thank you for your comments. We have made the necessary changes in the manuscript

2 In general, The UCB-MC-VEGF-FGF2-SDF1α group don’t have a proper control group. At least SDF1 alone and FGF alone should be added to the design of the study?

-In the same perspective, the authors should better explain the choice of groups. Why compare VEGF-FGF2-SDF1 cells to EGFP cells?

Response: Thank you very much for your comment. In our study we   tried evaluate the angiogenic effect of UCB-MCs modified with different adenoviruses  alone and in various combinations. We didn`t have a purpose to estimate the effect of viral transduction itself. If we focused on the above mentioned goal then we could consider non-treated UCB-MCs as a negative control.  In the in vivo block for UCB-MC-VEGF-FGF2-SDF1α, Ad-EGFP-modified cells served as controls, as well as fragments of the matrix without cells (PBS). Comparing VEGF-FGF2-SDF1 cells to EGFP cells, we evaluated angiogenic effect of growth factors.

Response 2: Thank you very much for your comment.

In our in vitro studies, the controls for UCB-MC-VEGF-FGF2-SDF1α are either Ad-EGFP-modified cells or unmodified cells. Our goal was not to analyze each type of modification in detail. In the in vivo block for UCB-MC-VEGF-FGF2-SDF1α, the controls were Ad-EGFP-modified cells, as well as fragments of the matrix without cells (PBS). Comparing VEGF-FGF2-SDF1 cells to EGFP cells, so we evaluate the effect of genetic modification.

Ad-EGFP is not a suitable control for VEGF-FGF2-SDF1α. Comparing VEGF-FGF2-SDF1 cells to EGFP cells don’t allow you to evaluate the effect of genetic modification, this is why proper controls are seriously lacking. In addition to adding the missing control groups, if the authors want to keep the EGFP comparison with VEGF-FGF2-SDF1α, these choices should be better discussed in the introduction and discussion.

Response: We have made the necessary changes in the manuscript.

It is also worth emphasizing that the VEGF-FGF2-SDF1 cells and VEGF cells did not differ from EGFP cells and NTC cells in vitro studies. What can be seen from the data of morphological and phenotypic studies and the profile of secreted factors (other than the expression of recombinant proteins). EGFP is therefore a perfect control in vitro as well as in vitro study in this regard.

Table 3.

  1. The cytokine profile would require more donors giving the high variability

Response 3: You may be right; at the same time, the number of donors is adequate in the context of the study design. In this study, isolated and characterized cells were used in subsequent stages of the research. Therefore, the viewed therapeutic effect was determined by these cells. Here, the biological/therapeutic effect is determined by the secretion of recombinant proteins by cells. As well, these studies were previously conducted by us, which showed changes in the expression of only recombinant proteins.

Could the authors provide de statistical analysis? Given the values reported in the table and especially the SD values, we do not find any statistical differences. The extreme variability does not allow to conclude a priori. Even without statistics, how do you explain that the secretion of UCB-MC-VEGF is more than 3 times higher than for UCB-MC-VEGF-FGF2-SDF1α.

Response: All groups were subject to statistical analysis. Statistical analysis was carried out using descriptive and analytical statistics.  This block contains the processed data. The individual characteristics of blood donors can explain variability. The unique features of donors can explain variability. In the group, one of the pools showed increased expression of VEGF. It can be assumed that the cells had a more significant number of transcription and translation factors, which accounted for one CMV promotor in UCB-MC-VEGF. Lack of competition for factors responsible for transgene expression.

  1. The authors discuss the adenovirus treatment effect with activation of secretion of cytokines by activation of the immune system. However, the control group is non treated. How the authors explain this discrepancy?

Response 4: In this case, the control group is UCB-MC-EGFP. We are evaluating the expression of therapeutic proteins and the effects of genetic modification on the secreted factor profile. Additionally, we are considering the natural secretome profile of the cells in our analysis. Our experiments have shown no immune response activation, such as the synthesis of cytokines and chemokines, due to genetic modification.

UCB-MC-EGFP is not a proper control but OK for the point 4.

  1. The Ad genetic modification induces a dramatic variability of the protein expression targeted by the genetic modification. How authors explain this? This should be discussed.

Response 5: In this case, this is the expected result. We are trying to achieve this effect through genetic modification. The modification UCB-MCs with Ads leads to the overexpression of recombinant proteins (growth factors - VEGF, FGF2, SDF1). At the same time, the profile of the other investigated factors is not affected.

The problem is not inter-group variability but intra-group variability. SD values are sometimes at the same level or even greater than the mean. It seems difficult to discuss these results given the statistical flaws.

Response: Cells are diverse due to the individual variability of donors. A trial was used that showed differences only for three proteins that are expression products of recombinant genes. All experiments were made in triplicates. In our study statistically significant changes were considered at p<0.05.

Figure 3.

  1. The controls previously suggested are missing here. As mentioned by authors, SDF1 is able to recruit EPCs or other cell types that can support angiogenesis. Consequently, to better understand the mechanisms, controls groups should be added to the design of the study

Response 6:  We agree with your comments and have made the necessary changes. In this study, the therapeutic agent is UCB-MC-VEGF-FGF2-SDF1α, so that we will focus on this group. The therapeutic strategy implemented involves using UCB-MC modified with three Ads (Ad-VEGF, Ad-FGF2, Ad-SDF1α). The control groups in this block are UCB-MC-EGFP – umbilical cord blood mononuclear cells modified with Ad-EGFP and PBS – Matrigel containing PBS. Of course, the consideration of groups, Ad-FGF2, Ad-SDF1α and their combinations is obvious. However, studies of this plan were carried out by other research groups and us earlier.

You need to show the advantage of genetically modified cells. In order to do this and show that UCB-MC-VEGF-FGF2-SDF1α is a therapeutic strategy that should be considered, you must use the right controls. Again, UCB-MC-EGFP is not a proper control. If such studies have been made, they should be introduced in the manuscript not only as a citation for material and methods. You must clarify the identification and choice of your therapeutic strategy.

Response: Unfortunately, we are not able to change this block at the moment. Yes, it was an advised decision based on our previous studies and common protocols of the scientific community. 

  1. The Matrigel-plug experiment lack of an important control group that is strictly mandatory: non treated UCB-MC

Response 7: In this case, the untreated UCB-MC group is represented by the UCB-MC-EGFP group. Moreover, this group characterizes the absence of the effect of genetic modification on UCB-MC. In particular, we previously showed that the genetic modification of cells (Ad-EGFP) does not affect the profile of secreted factors (https://doi.org/10.1182/blood-2020-138664, https://doi.org/10.3390/ijms21186858). Therefore, this control group is sufficient for our study.

We appreciate your feedback and will consider adding additional control groups in future studies.

Again, UCB-MC-EGFP is not a proper control. If this paper will be published as it is. The authors need to clarify the choice of their control and pointing out the weakness of choosing such unappropriated control. The authors need to clarify the design of the study. When you designed the study you really chosen to have EGFP group as a control group? 

Response: Unfortunately, we are not able to change this block at the moment. We believe the EGFP group is a good and adequate control group in the present study.

  1.  Could the authors provide picture used for microvascular density? What ab was used? It would be interesting to use an anti-human and an anti-mouse antibodies in order to differentiate between the vasculogenesis occurring in the plug and the cell recruitment.

Response 8: The use of specific anti-human and anti-mouse antibodies certainly has a potential scientific interest. Visualization of microvascular density was performed without the use of antibodies. The assessment of the formed vessels was carried out visually with hematoxylin & eosin (H&E) staining. However, it is also worth considering that the formed vessels are derived from the mouse component. Transplanted cells secrete growth and trophic factors (paracrine stimulation) and recruit recipient cells.

The authors could add another method to quantify the microvascular density. Vessels should be properly stained. HE pictures are not representative of the quantifications (VEGF and EGFP). These elements about cell recruitment and the origin of the endothelial cells giving raise to the vessel within the plug, should be at least discussed in the manuscript if any experiments are going to be run.

Response: Unfortunately, we are not able to change this block at the moment and add another method to quantify the microvascular density. We agree that immunological staining using specific antibodies ,f.e. to  leptin, is a reliable technique for evaluation of newly formed vessels. However, HE staining remains the well-known and approved standard to estimate a)morphological changes; b)vessel formation; c)cell origin and shouldn`t be neglected.

  1. Again, why EGFP compared to the other 2 groups? the scope is ambiguous here.

Response 9: 

In this case, using EGFP as a control group is appropriate for our study. In the study, we assess the impact of genetic modification. Native cells (in our case, AD-EGFP-modified cells, which do not differ from unmodified (NTC) cells) can exercise a pro-angiogenic effect by secreting various proactive molecules.,

By using EGFP as a control you don’t assess the impact of genetic modification. You assess the difference between modifying EGFP or a cocktail of growth factors. EGFP modified cells are not native as obviously mentioned by their name. The authors should explain better the design of the study. Why the control group wasn’t added? 

Response: Thank you very much for your observations. Earlier in this investigation, we have shown that genetic modification does not affect AD-EGFP on cells' morphological and functional properties. We also consider that AD-EGFP is identical to unmodified (NTC). We agree that comparison between various types of control groups is a golden standard and an ideal strategy for confirmation of hypothesis. As it was mentioned above the main idea of the study was to demonstrate the angiogenic effect of growth factors carried by adenoviral vectors. Reporter genes (EGFP, RFP, LUC, YFP) are well known examples of positive controls that normally doesn`t effect on cell state and cytokine release while using in appropriate concentrations. And in the current work viral load for the cells was equal so that Ad5-EGFP- treated UCB-MCs could be considered better control than non-treated cells.

Figure 4.

  1. How authors observed the recruited fibroblasts in the plugs?

Response 10: Recruitment of cells was assessed by imaging histological sections and by counting migrating fibroblasts

This is not answering the question? How were you able to identify the fibroblasts?

Response: The cells were not stained, and the evaluation was carried out only visually and morphologically. Fibroblast like cells distinguish with its size and morphology so that we could evaluate its migration

  1. It looks like there are post-fixation issues in some of the pictures. Are the authors confident with the staining?

Response 11: Thank you for your comment. The authors are confident in the results presented. Additionally, histological data is consistent with the results of the RT-qPCR analysis.

Could the authors provide other pictures? considering the pictures presented it means that there is high secretion of VEGF in the plugs injected with only PBS?

Response: Necessary changes made.

  1. Could the authors explain better the color code for the staining.

Response 12: Necessary changes made.

OK

  1. Other angiogenesis assays should be added to reinforce the manuscript. At least in vitro experiments that would also contribute to the characterization of the cells.

Response 13: Thank you very much for your comment. In previous research, we conducted whole transcriptome profiling and demonstrated the impact of supernatants collected from genetically modified cells on HUVEC (in vitro) proliferation. We have also shown the ability of modified UCB to differentiate into different cell types and stimulate angiogenesis in various in vivo models (https://doi.org/10.3390/ijms21186858, https://doi.org/10.1007/s10517-013-1999-2,https://doi.org/10.1007/s11055-013-9778-0, https://doi.org/10.1007/s12035-016-0017-x, https://doi.org/10.1038/sc.2015.162, https://doi.org/10.3390/cells10020432).

This should be added to the manuscript. However, none of these paper show the study of the genetic modifications presented in angiogenesis assays such as tubulogenesis assays, 3D sprouting, wound healing… 

Response: Thanks for the suggestion. In this study, it was not the focus of our interest. We will implement them in future works.

Minor concerns:

  1. MM table 2: Is there a mistake in the ab list? The secondaries are all anti-rabbit?

Response 14: Thank you for your comment. The necessary changes have been made.

OK

  1. Microvessel density analysis material and methods are missing

Response 15: Thank you for your comment. The necessary changes have been made.

How vessels were identified? with presence of erythrocytes in the lumen? 

Response: Vessels were identified and counted after histological analysis.

  1. References about state-of-the-art methods to study angiogenesis should be added

Response 16: Thank you very much for your comment. Many methods are currently available to study angiogenesis (https://doi.org/10.3390/biomedicines7020037, https://doi.org/10.1155/2020/8857428). In the current article, we focused on one is the most classical methods for angiogenesis analysis.

That's nice but I wasn't speaking for myself, it was a suggestion to add to your manuscript.

Response: Thank you very much, we already cited these papers in the current manuscript.

Round 3

Reviewer 2 Report

Round 2 

Comments in the Author's Notes in purple

Round 3

Comments in green after Author’s response 

Major concerns:
1. Figure 1: after the determination of the phenotype of UCB-MC used, the authors should perform the same experiment in the genetically modified cells in order to understand if the genetic modification leads to the differentiation of cells. ECFC markers and endothelial cell markers should be looked at. This would really strengthen the manuscript as a proper characterization of the modified cells should be performed.
Response 1: Thank you very much for your comments.
The manuscript has been revised. In this study, we did not focus on the characterization of the isolated cells. In the study, we showed the genetically modified cells had the same immunophenotype as the unmodified cells. The study showed that the genetically modified cells had the same immunophenotype as the unmodified cells. Moreover, in previously published research, we have shown that genetic modification does not affect the mRNA landscape of the cells. Furthermore, we have shown in previous work that genetic modifications do not alter the reorganization of translated mRNAs (transcriptome) in transduced cells. We also did not find any differences in the mRNA expression profile of ECFC CD-markers after cell modification with recombinant adenoviruses (https://doi.org/10.1182/blood- 2020-138664). 

I am sorry, the figure 1 seems to show the phenotype of UCB-MCs, which is already known. To give an input to this figure the authors need to add the same experiments but in the genetically modified cells. It means showing results of the experiments with figures and not only adding a sentence in the results section. If the authors
Response: Thank you for your comments. We have made the necessary changes in the manuscript 

Which genetic modification has been made to the “modified CD34+ cells” figures? The authors should present the phenotype for each modification presented in the paper. 

2 In general, The UCB-MC-VEGF-FGF2-SDF1α group don’t have a proper control group. At least SDF1 alone and FGF alone should be added to the design of the study?
-In the same perspective, the authors should better explain the choice of groups. Why compare VEGF-FGF2-SDF1 cells to EGFP cells? 

Response: Thank you very much for your comment. In our study we tried evaluate the angiogenic effect of UCB- MCs modified with different adenoviruses alone and in various combinations. We didn`t have a purpose to estimate the effect of viral transduction itself. If we focused on the above mentioned goal then we could consider non-treated UCB-MCs as a negative control. In the in vivo block for UCB-MC-VEGF-FGF2-SDF1α, Ad-EGFP-modified cells served as controls, as well as fragments of the matrix without cells (PBS). Comparing VEGF-FGF2-SDF1 cells to EGFP cells, we evaluated angiogenic effect of growth factors. 

Response 2: Thank you very much for your comment.
In our in vitro studies, the controls for UCB-MC-VEGF-FGF2-SDF1
α are either Ad-EGFP-modified cells or unmodified cells. Our goal was not to analyze each type of modification in detail. In the in vivo block for UCB-MC- VEGF-FGF2-SDF1α, the controls were Ad-EGFP-modified cells, as well as fragments of the matrix without cells (PBS). Comparing VEGF-FGF2-SDF1 cells to EGFP cells, so we evaluate the effect of genetic modification. 

Ad-EGFP is not a suitable control for VEGF-FGF2-SDF1α. Comparing VEGF-FGF2-SDF1 cells to EGFP cells don’t allow you to evaluate the effect of genetic modification, this is why proper controls are seriously lacking. In addition to adding the missing control groups, if the authors want to keep the EGFP comparison with VEGF-FGF2- SDF1α, these choices should be better discussed in the introduction and discussion. 

Response: We have made the necessary changes in the manuscript.
It is also worth emphasizing that the VEGF-FGF2-SDF1 cells and VEGF cells did not differ from EGFP cells and NTC cells in vitro studies. What can be seen from the data of morphological and phenotypic studies and the profile of secreted factors (other than the expression of recombinant proteins). EGFP is therefore a perfect control in vitro 

as well as in vitro study in this regard. 

Table 3. 

Please define the “in vitro studies”

Where can we find the data of morphological and phenotypic studies ? 

3. The cytokine profile would require more donors giving the high variability 

Response 3: You may be right; at the same time, the number of donors is adequate in the context of the study design. In this study, isolated and characterized cells were used in subsequent stages of the research. Therefore, the viewed therapeutic effect was determined by these cells. Here, the biological/therapeutic effect is determined by the secretion of recombinant proteins by cells. As well, these studies were previously conducted by us, which showed changes in the expression of only recombinant proteins. 

Could the authors provide de statistical analysis? Given the values reported in the table and especially the SD values, we do not find any statistical differences. The extreme variability does not allow to conclude a priori. Even without statistics, how do you explain that the secretion of UCB-MC-VEGF is more than 3 times higher than for UCB-MC-VEGF-FGF2-SDF1α

Response: All groups were subject to statistical analysis. Statistical analysis was carried out using descriptive and analytical statistics. This block contains the processed data. The individual characteristics of blood donors can explain variability. The unique features of donors can explain variability. In the group, one of the pools showed increased expression of VEGF. It can be assumed that the cells had a more significant number of transcription and translation factors, which accounted for one CMV promotor in UCB-MC-VEGF. Lack of competition for factors responsible for transgene expression. 

Could the authors provide the statistical test that was run and show the raw data? 

If only one of the pools showed increased expression, does it mean that the transfection is not working every time? How does the authors select the cell used for further assays? 

4. The authors discuss the adenovirus treatment effect with activation of secretion of cytokines by activation of the immune system. However, the control group is non treated. How the authors explain this discrepancy? 

Response 4: In this case, the control group is UCB-MC-EGFP. We are evaluating the expression of therapeutic proteins and the effects of genetic modification on the secreted factor profile. Additionally, we are considering the natural secretome profile of the cells in our analysis. Our experiments have shown no immune response activation, such as the synthesis of cytokines and chemokines, due to genetic modification. 

UCB-MC-EGFP is not a proper control but OK for the point 4. 

5. The Ad genetic modification induces a dramatic variability of the protein expression targeted by the genetic modification. How authors explain this? This should be discussed. 

Response 5: In this case, this is the expected result. We are trying to achieve this effect through genetic modification. The modification UCB-MCs with Ads leads to the overexpression of recombinant proteins (growth factors - VEGF, FGF2, SDF1). At the same time, the profile of the other investigated factors is not affected. 

The problem is not inter-group variability but intra-group variability. SD values are sometimes at the same level or even greater than the mean. It seems difficult to discuss these results given the statistical flaws. 

ResponseCells are diverse due to the individual variability of donors. A trial was used that showed differences only for three proteins that are expression products of recombinant genes. All experiments were made in triplicates. In our study statistically significant changes were considered at p<0.05. 

Please provide the statistical test that was run and show the raw data.

Round 3

Comments in green after Author’s response 

Major concerns:
1. Figure 1: after the determination of the phenotype of UCB-MC used, the authors should perform the same experiment in the genetically modified cells in order to understand if the genetic modification leads to the differentiation of cells. ECFC markers and endothelial cell markers should be looked at. This would really strengthen the manuscript as a proper characterization of the modified cells should be performed.
Response 1: Thank you very much for your comments.
The manuscript has been revised. In this study, we did not focus on the characterization of the isolated cells. In the study, we showed the genetically modified cells had the same immunophenotype as the unmodified cells. The study showed that the genetically modified cells had the same immunophenotype as the unmodified cells. Moreover, in previously published research, we have shown that genetic modification does not affect the mRNA landscape of the cells. Furthermore, we have shown in previous work that genetic modifications do not alter the reorganization of translated mRNAs (transcriptome) in transduced cells. We also did not find any differences in the mRNA expression profile of ECFC CD-markers after cell modification with recombinant adenoviruses (https://doi.org/10.1182/blood- 2020-138664). 

I am sorry, the figure 1 seems to show the phenotype of UCB-MCs, which is already known. To give an input to this figure the authors need to add the same experiments but in the genetically modified cells. It means showing results of the experiments with figures and not only adding a sentence in the results section. If the authors
Response: Thank you for your comments. We have made the necessary changes in the manuscript 

Which genetic modification has been made to the “modified CD34+ cells” figures? The authors should present the phenotype for each modification presented in the paper. 

Response: Thank you for your comments and critical reading of the manuscript. We have made the necessary changes in the manuscript. In the study, we focused on the more important therapeutic group UCB-MC-VEGF-FGF2-SDF1α, which has been comprehensively characterized in the current study.

2 In general, The UCB-MC-VEGF-FGF2-SDF1α group don’t have a proper control group. At least SDF1 alone and FGF alone should be added to the design of the study?
-In the same perspective, the authors should better explain the choice of groups. Why compare VEGF-FGF2-SDF1 cells to EGFP cells? 

Response: Thank you very much for your comment. In our study we tried evaluate the angiogenic effect of UCB- MCs modified with different adenoviruses alone and in various combinations. We didn`t have a purpose to estimate the effect of viral transduction itself. If we focused on the above mentioned goal then we could consider non-treated UCB-MCs as a negative control. In the in vivo block for UCB-MC-VEGF-FGF2-SDF1α, Ad-EGFP-modified cells served as controls, as well as fragments of the matrix without cells (PBS). Comparing VEGF-FGF2-SDF1 cells to EGFP cells, we evaluated angiogenic effect of growth factors. 

Response 2: Thank you very much for your comment.
In our in vitro studies, the controls for UCB-MC-VEGF-FGF2-SDF1α are either Ad-EGFP-modified cells or unmodified cells. Our goal was not to analyze each type of modification in detail. In the in vivo block for UCB-MC- VEGF-FGF2-SDF1α, the controls were Ad-EGFP-modified cells, as well as fragments of the matrix without cells (PBS). Comparing VEGF-FGF2-SDF1 cells to EGFP cells, so we evaluate the effect of genetic modification. 

Ad-EGFP is not a suitable control for VEGF-FGF2-SDF1α. Comparing VEGF-FGF2-SDF1 cells to EGFP cells don’t allow you to evaluate the effect of genetic modification, this is why proper controls are seriously lacking. In addition to adding the missing control groups, if the authors want to keep the EGFP comparison with VEGF-FGF2- SDF1α, these choices should be better discussed in the introduction and discussion. 

Response: We have made the necessary changes in the manuscript.
It is also worth emphasizing that the VEGF-FGF2-SDF1 cells and VEGF cells did not differ from EGFP cells and NTC cells in vitro studies. What can be seen from the data of morphological and phenotypic studies and the profile of secreted factors (other than the expression of recombinant proteins). EGFP is therefore a perfect control in vitro 

as well as in vitro study in this regard. 

Table 3. 

Please define the “in vitro studies”

Where can we find the data of morphological and phenotypic studies ? 

Response: «In vitro is Latin for “in glass.” It describes procedures, tests, and experiments that researchers perform outside of a living organism» (https://www.medicalnewstoday.com/articles/in-vivo-vs-in-vitro).

Figures 1 and 2 present morphological and phenotypic characterization of cells.

  1. The cytokine profile would require more donors giving the high variability 

Response 3: You may be right; at the same time, the number of donors is adequate in the context of the study design. In this study, isolated and characterized cells were used in subsequent stages of the research. Therefore, the viewed therapeutic effect was determined by these cells. Here, the biological/therapeutic effect is determined by the secretion of recombinant proteins by cells. As well, these studies were previously conducted by us, which showed changes in the expression of only recombinant proteins. 

Could the authors provide de statistical analysis? Given the values reported in the table and especially the SD values, we do not find any statistical differences. The extreme variability does not allow to conclude a priori. Even without statistics, how do you explain that the secretion of UCB-MC-VEGF is more than 3 times higher than for UCB-MC-VEGF-FGF2-SDF1α. 

Response: All groups were subject to statistical analysis. Statistical analysis was carried out using descriptive and analytical statistics. This block contains the processed data. The individual characteristics of blood donors can explain variability. The unique features of donors can explain variability. In the group, one of the pools showed increased expression of VEGF. It can be assumed that the cells had a more significant number of transcription and translation factors, which accounted for one CMV promotor in UCB-MC-VEGF. Lack of competition for factors responsible for transgene expression. 

Could the authors provide the statistical test that was run and show the raw data? 

If only one of the pools showed increased expression, does it mean that the transfection is not working every time? How does the authors select the cell used for further assays? 

Response: Thank you very much for your comment.

We have made the necessary changes in the manuscript. We conducted additional studies to improve the reliability of the data presented. Additionally, we doubled the sample size. Our results show that transduction works every time.

  1. The authors discuss the adenovirus treatment effect with activation of secretion of cytokines by activation of the immune system. However, the control group is non treated. How the authors explain this discrepancy? 

Response 4: In this case, the control group is UCB-MC-EGFP. We are evaluating the expression of therapeutic proteins and the effects of genetic modification on the secreted factor profile. Additionally, we are considering the natural secretome profile of the cells in our analysis. Our experiments have shown no immune response activation, such as the synthesis of cytokines and chemokines, due to genetic modification. 

UCB-MC-EGFP is not a proper control but OK for the point 4. 

  1. The Ad genetic modification induces a dramatic variability of the protein expression targeted by the genetic modification. How authors explain this? This should be discussed. 

Response 5: In this case, this is the expected result. We are trying to achieve this effect through genetic modification. The modification UCB-MCs with Ads leads to the overexpression of recombinant proteins (growth factors - VEGF, FGF2, SDF1). At the same time, the profile of the other investigated factors is not affected. 

The problem is not inter-group variability but intra-group variability. SD values are sometimes at the same level or even greater than the mean. It seems difficult to discuss these results given the statistical flaws. 

Response: Cells are diverse due to the individual variability of donors. A trial was used that showed differences only for three proteins that are expression products of recombinant genes. All experiments were made in triplicates. In our study statistically significant changes were considered at p<0.05. 

Please provide the statistical test that was run and show the raw data.

Response: We have made the necessary changes in the manuscript
